# Mitigating Manifold Departure: Uncertainty-Aware Subspace Rectification for Trustworthy MLLM Decoding

Yingxuan Zhuang [1]   Jingxiao Yang [1]   Miao Pan [1]   Cheng Tan [2]   Yuxiang Cai [1]   Siwei Tan [1]   Chen Zhi [1]
Xuhong Zhang [1]   Jianwei Yin [1]   Jintao Chen [1][3]

## Abstract

MLLMs frequently hallucinate objects inconsistent with visual inputs. This issue is typically attributed to the over-reliance on language priors, which can override the visual context. Recent training-free decoding strategies address this by penalizing language priors. However, these methods overlook the dual nature of language priors, where they can be both helpful and harmful depending on the alignment with visual evidence. In particular, blindly suppressing language priors often disrupts the model's semantic manifold, leading to performance degradation, a phenomenon we term Manifold Departure. To address this, we propose Manifold-Guided Adaptive Projection (MGAP), a geometry-aware, training-free decoding method that mitigates hallucinations while preserving representation structure. MGAP first constructs a language-prior subspace from blind hidden states via SVD. During decoding, MGAP projects each multimodal hidden state onto this subspace and applies a consistency-aware gate to adaptively attenuate only the projected prior component, yielding a subspace-selective update that largely preserves the orthogonal semantic components. Extensive experiments on POPE and CHAIR show that MGAP outperforms prior decoding baselines, achieving stronger hallucination suppression without sacrificing coherence.

## 1. Introduction

In recent years, Multimodal Large Language Models (MLLMs) (Wang et al., 2024b; Chen et al., 2024a; Deitke

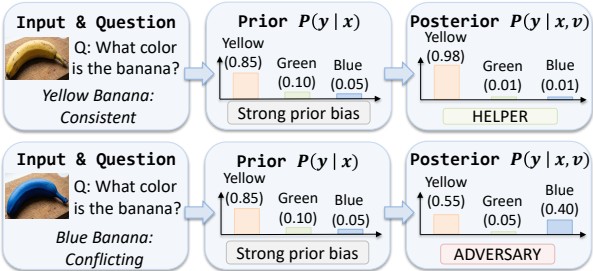

*Figure 1.* The dual role of language priors in MLLM decoding. When visual evidence and priors are aligned (yellow banana), priors sharpen and stabilize generation. When they conflict (blue banana), priors can override the image and induce hallucination.

et al., 2024) have become strong general-purpose interfaces for vision and language, enabling multimodal reasoning and generation across diverse benchmarks (Yue et al., 2024; Liu et al., 2024b; Hu et al., 2025). Despite this progress, they remain vulnerable to object hallucination (Li et al., 2023c; Bai et al., 2024; Rohrbach et al., 2018): the model produces fluent, plausible content that is simply not supported by the image. This failure is often traced to the model's language priors learned during pre-training, which can overpower weak or ambiguous visual evidence (Leng et al., 2024; Huang et al., 2024; Li et al., 2023a; Tong et al., 2024). Motivated by this, a growing line of training-free decoding methods (Leng et al., 2024; Favero et al., 2024; Chen et al., 2024b; Wang et al., 2024c; Park et al., 2025) attempts to mitigate hallucinations by subtracting a bias term (e.g., an unconditional branch or a contrastive context) from the decoding logits. However, the core difficulty is that language priors are not uniformly harmful. As illustrated in Figure 1, priors play a dual role: they can trigger hallucinations under vision–language conflict (blue banana), yet serve as confidence anchors when aligned with the image (yellow banana), sharpening and stabilizing generation. This makes indiscriminate prior suppression a blunt instrument: it may remove exactly the structure that helps normal, visually consistent cases.

We further show that this unintended degradation has a geo-

---

[1]Ningbo Global Innovation Center, Zhejiang University [2]Shanghai Artificial Intelligence Laboratory [3]Zhejiang Key Laboratory of Digital-Intelligence Service Technology. Correspondence to: Jintao Chen <chenjintao@zju.edu.cn>.

*Proceedings of the $43^{rd}$ International Conference on Machine Learning*, Seoul, South Korea. PMLR 306, 2026. Copyright 2026 by the author(s).

metric signature. Viewed in representation space, valid decoding trajectories concentrate in a structured high-density region (Bengio et al., 2013a; Verma et al., 2019; Stutz et al., 2019; Khoury & Hadfield-Menell, 2018). Linear suppression methods apply a global, structure-agnostic shift, which can push hidden states toward low-density regions and destabilize decoding. We term this failure mode *Manifold Departure* and analyze it in Section 3.3 (Figure 3). To resolve this tension, we propose Manifold-Guided Adaptive Projection (MGAP), a geometry-aware, training-free decoding framework. MGAP explicitly models language priors as a low-rank prior subspace estimated from blind hidden states via SVD, and performs a selective intervention during decoding: each hidden state is projected onto the prior subspace, and only this projected component is adaptively attenuated when vision–language conflict is detected.

In summary, our main contributions are:

- We identify Manifold Departure as a geometric failure mode of linear, training-free prior suppression, explaining why such methods can degrade performance even when priors align with vision.

- We propose MGAP, which learns a language-prior subspace from blind states and applies *adaptive, subspace-selective projection* to suppress hallucinations without disturbing orthogonal semantics.

- Experiments on POPE (Li et al., 2023c) and CHAIR (Rohrbach et al., 2018) show that MGAP achieves stronger trade-offs between hallucination mitigation and descriptive fidelity than prior training-free decoding baselines.

## 2. Related Work

**Hallucination in MLLMs** Object hallucination is a long-standing issue in vision–language generation and is amplified in modern MLLMs. Surveys summarize its phenomena and taxonomies (Bai et al., 2024), while benchmarks/protocols quantify object-level ungrounded content (Li et al., 2023c;b). Similar failures are also studied in other generation settings such as neural machine translation (Guerreiro et al., 2023). Training-time mitigation (e.g., robust instruction tuning, factual alignment, preference optimization) can reduce hallucinations but requires extra supervision or optimization (Liu et al., 2024a; Sun et al., 2023; Zhao et al., 2023; Jiang et al., 2024). We instead study training-free, decoding-time control.

**Inference-time intervention** Decoding-time methods mitigate hallucination without parameter updates. VCD suppresses language priors by contrasting multimodal decoding against a biased/unconditioned branch (Leng et al.,

2024). This idea is closely related to contrastive decoding in text generation (Li et al., 2023a) and has been further explored in subsequent VCD-style analyses (Lee et al., 2024). OPERA regulates over-trust via penalty and retrospection-allocation (Huang et al., 2024). Other approaches include instruction contrastive decoding (Wang et al., 2024c), adaptive focal contrast decoding (Chen et al., 2024b), and introspective/ensemble decoding (Huo et al., 2025; Cho et al., 2025; Park et al., 2025). Many can be seen as global, structure-agnostic linear manipulations in representation/logit space, which may hurt outputs even when priors align with vision. Our method instead models a prior subspace and intervenes selectively in a geometry-aware manner. We additionally compare MGAP against recent decoding-time methods including DeCo (Wang et al., 2024a) and MoD (Su et al., 2025). MGAP consistently achieves the best overall trade-off between hallucination mitigation and generation fidelity across both backbones.

**Manifold structure of latent representations** Representation geometry is widely used to explain and control model behavior via low-dimensional structure, dominant subspaces, and concentration near a data manifold induced by the training distribution. Following the common manifold assumption in representation learning, high-dimensional observations are often assumed to lie on or near a smooth low-dimensional manifold (Roweis & Saul, 2000; Tenenbaum et al., 2000; Bengio et al., 2013b). In vision, transformer representations under large-scale pretraining exhibit strong geometric regularities (Dosovitskiy et al., 2020; He et al., 2022; Beyer et al., 2023). Related work measures cross-layer/model alignment using SVCCA/CKA-style diagnostics (Raghu et al., 2017; Kornblith et al., 2019; Morcos et al., 2018; Li et al., 2018). Decoding-time behavior in language generation has also been connected to representation-level objectives and contrastive directions (Li et al., 2023a). Building on this view, we adopt the same geometric perspective for MLLM hidden states: we estimate a low-rank language-prior subspace from blind hidden states and apply bounded, selective projection to mitigate hallucination while staying near the data manifold.

## 3. Preliminary

In this section, we introduce notation for multimodal decoding and summarize training-free linear suppression under a unified form. We then clarify our manifold viewpoint that motivates geometry-aware decoding interventions.

### 3.1. Representation-Level View of Training-Free Decoding

Given an image $v$ and a textual query $x$, a multimodal large language model generates tokens autoregressively. At de-

coding step $t$, the next-token distribution is given by

$$P(y_t \mid y_{<t}, x, v) = \text{Softmax}(\mathcal{F}(h_t)), \quad (1)$$

where $h_t \in \mathbb{R}^d$ denotes the final-layer hidden state.

Recent training-free hallucination mitigation methods intervene at decoding time by linearly suppressing language priors. Despite different formulations, many such methods (Leng et al., 2024; Wang et al., 2024c; Kim et al., 2024) can be unified as

$$\text{Logits}_{\text{final}} = \text{Logits}_{\text{main}} - \rho \cdot \text{Logits}_{\text{bias}}, \quad (2)$$

where $\text{Logits}_{\text{bias}}$ is obtained from a biased context (e.g., blind or perturbed inputs).

Importantly, Eq. (2) implicitly assumes that hallucinations can be mitigated by a global linear translation in representation space. In this work, we analyze and intervene directly on the hidden representation $h_t$, enabling a geometric understanding of why such linear suppression may fail.

### 3.2. Hidden-State Manifold and Manifold Departure

Let $h_t \in \mathbb{R}^d$ denote the (last-layer) decoder hidden state at decoding step $t$. Following the common manifold assumption in representation learning, high-dimensional observations are often assumed to lie on or near a smooth low-dimensional manifold (Roweis & Saul, 2000; Tenenbaum et al., 2000; Bengio et al., 2013b). We adopt the same view for MLLM hidden states.

**Definition 3.1** (Hidden-state manifold). Let $P_0$ denote the distribution of hidden states produced by *normal decoding* (without any decoding-time intervention). We say the hidden states concentrate near a $r$-dimensional manifold $\mathcal{M} \subset \mathbb{R}^d$ ($r \ll d$) if there exist $\varepsilon > 0$ and a small $\delta \in (0, 1)$ such that:

$$\Pr_{h \sim P_0} \left[ \text{dist}(h, \mathcal{M}) \leq \varepsilon \right] \geq 1 - \delta. \quad (3)$$

Here $\Pr_{h \sim P_0}[\cdot]$ denotes probability when $h$ is sampled from $P_0$. Eq. (3) means that at least a $(1 - \delta)$ fraction of normal-decoding states lie within an $\varepsilon$-neighborhood (an "$\varepsilon$-tube") around $\mathcal{M}$. We define the Euclidean distance to the manifold as:

$$\text{dist}(h, \mathcal{M}) \triangleq \inf_{m \in \mathcal{M}} \|h - m\|_2. \quad (4)$$

We refer to $\mathcal{M}$ as the *semantic manifold* of normal decoding.

**A computable proxy** Since $\mathcal{M}$ and $\text{dist}(h, \mathcal{M})$ are not directly computable, we approximate them using a reference bank of hidden states $\mathcal{S} = \{h^{(i)}\}_{i=1}^N$ collected from normal decoding (gray points in Fig. 3). For any query state $h \in \mathbb{R}^d$, let $\text{NN}_k(h; \mathcal{S}) \subset \mathcal{S}$ denote the set of its $k$ nearest neighbors in $\mathcal{S}$ under the chosen metric (default: Euclidean distance).

We define an off-manifoldness score by the average $k$NN distance:

$$d_k(h; \mathcal{S}) \triangleq \frac{1}{k} \sum_{s \in \text{NN}_k(h; \mathcal{S})} \|h - s\|_2. \quad (5)$$

(Using cosine distance after $\ell_2$-normalization is an alternative in high dimensions.)

**Definition 3.2** (Manifold departure). Consider an intervention $T$ applied at step $t$ that maps $h_t$ to $\tilde{h}_t = T(h_t)$. We say $T$ induces *manifold departure* at step $t$ if:

$$d_k(\tilde{h}_t; \mathcal{S}) > \tau, \quad (6)$$

where the threshold $\tau$ is set as the $(1 - \delta)$-quantile of the reference scores $\{d_k(s; \mathcal{S})\}_{s \in \mathcal{S}}$, i.e., only an $\delta$ fraction of normal-decoding states would be labeled off-manifold under this criterion.

### 3.3. Manifold Departure from Prior Suppression

We identify a consistent *failure mechanism* underlying many training-free linear decoding interventions: *indiscriminate* suppression of language-prior components can push hidden states away from the normal-decoding semantic manifold (Definition 3.1), thereby degrading generation quality even when the language prior is semantically helpful.

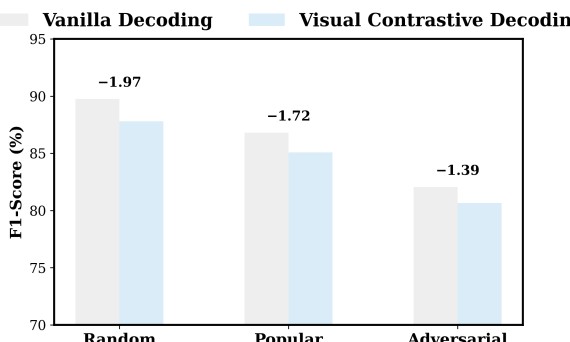

*Figure 2.* Uniform linear prior suppression can hurt. VCD (vs. vanilla) shows consistent performance drops on POPE across splits, including standard cases where priors align with visual evidence.

**Empirical observation** We compare vanilla decoding and Visual Contrastive Decoding (VCD) (Leng et al., 2024) on the POPE benchmark using LLaVA v1.5-7B. As shown in Figure 2, VCD consistently degrades performance across all evaluation splits. Notably, this degradation also occurs in standard cases where visual evidence and language priors are naturally aligned, suggesting a systematic limitation of applying the same linear suppression regardless of context.

**Geometric mechanism** To understand *why* such degradation happens, we analyze hidden representations through

the lens of the semantic manifold defined in Section 3.2. We focus on the final-layer hidden state $h_t \in \mathbb{R}^D$ and its evolution under decoding-time interventions.

We extract hidden states from normal decoding trajectories on the COCO dataset and visualize their distribution using Principal Component Analysis (PCA). As shown in Figure 3, hidden states corresponding to valid generations concentrate within a structured, low-dimensional region of the representation space, which we interpret as the normal-decoding semantic manifold (Definition 3.1). Linear extrapolation along the VCD direction, however, progressively moves states away from this region, as quantified by increasing off-manifoldness scores shown in the inset.

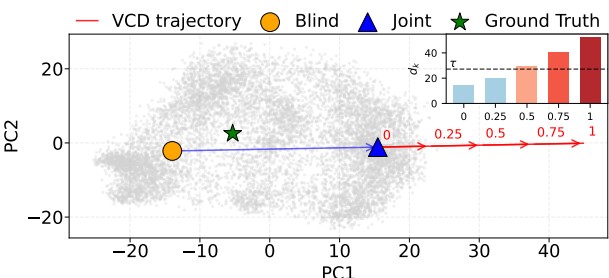

*Figure 3.* Manifold departure under linear suppression. Gray points are hidden states from normal decoding (semantic manifold); orange/blue/green mark blind prior, joint posterior, and ground-truth states. Red numeric labels denote different values of the VCD extrapolation coefficient $\alpha$. *Inset*: kNN-based off-manifoldness score $d_k$ of VCD states at different $\alpha$, computed in the original hidden space. The dashed line indicates the threshold $\tau$ used to define manifold departure (Eq. (6)); bars above $\tau$ are increasingly shaded to reflect larger deviations from the semantic manifold.

In contrast, linear prior suppression applies a fixed contrastive shift that extrapolates along the difference between the blind (prior) and joint (visual) contexts, e.g.,

$$h' = h_{\text{joint}} + \rho\big(h_{\text{joint}} - h_{\text{blind}}\big). \tag{7}$$

Here $h_{\text{blind}}$ follows VCD (Leng et al., 2024) and is obtained by running the model on a visually distorted input, where the original image is perturbed with diffusion noise to form the blind view. Such a global linear extrapolation is agnostic to the local geometry of the normal-decoding manifold. Even when $h_{\text{joint}}$ lies in a typical (high-density) region, stepping along the direction $(h_{\text{joint}} - h_{\text{blind}})$ can move the state into a tail region that is rarely visited under normal decoding. Formally, this corresponds to increased off-manifoldness (Eq. (5)) and can trigger the manifold-departure condition $d_k(h_t; \mathcal{S}) > \tau$ (Definition 3.2), which is visualized along the VCD trajectory in Figure 3 (inset). Once the hidden state departs from the high-density region supported by normal decoding, the decoder operates in a poorly supported regime, often leading to less stable token distributions and degraded generation quality (Fig 2).

Importantly, this failure arises from the structure-agnostic nature of uniform suppression, rather than from the intrinsic correctness or incorrectness of the language prior. This diagnosis motivates a selective and geometry-aware intervention, which we introduce next.

## 4. Method

### 4.1. Label-Free Language Prior Subspace Construction

Motivated by the diagnosis in Section 3.3, we model language priors as a low-dimensional subspace in the hidden-state space. We construct this subspace using only queries, requiring no labels, image content, or parameter updates.

Given prompts $\{x^{(i)}\}_{i=1}^N$, we collect final-layer blind hidden states $\{h_{\text{blind}}^{(i)} \in \mathbb{R}^d\}_{i=1}^N$. Let $\bar{h}_{\text{blind}} \triangleq \frac{1}{N}\sum_{i=1}^N h_{\text{blind}}^{(i)}$ and define the centered matrix

$$\tilde{H}_{\text{blind}} \triangleq \begin{bmatrix} (h_{\text{blind}}^{(1)} - \bar{h}_{\text{blind}})^\top \\ \vdots \\ (h_{\text{blind}}^{(N)} - \bar{h}_{\text{blind}})^\top \end{bmatrix} \in \mathbb{R}^{N \times d}. \tag{8}$$

We estimate the prior subspace as the top-$K$ principal components, i.e.,

$$V_{\text{prior}} \in \arg\max_{V \in \mathbb{R}^{d \times K}} \text{Tr}\Big(V^\top \tilde{H}_{\text{blind}}^\top \tilde{H}_{\text{blind}} V\Big). \tag{9}$$

Here $\text{Tr}(\cdot)$ denotes the trace (sum of diagonal entries); $\text{Tr}(\cdot)$ is the total variance of blind hidden states captured by the $K$ orthonormal directions $V_{\text{prior}} = [v_1, \ldots, v_K]$, which has the closed-form solution via SVD:

$$\tilde{H}_{\text{blind}} = U\Sigma V^\top, \qquad V_{\text{prior}} \triangleq V_{[:,1:K]} \in \mathbb{R}^{d \times K}. \tag{10}$$

$V_{\text{prior}}$ captures dominant blind-state variations induced by linguistic regularities; it is *not* assumed inherently harmful, motivating adaptive (rather than uniform) suppression in subsequent decoding.

### 4.2. Consistency-Aware Adaptive Projection

Given a hidden state $h_{\text{orig}}$ produced during standard multimodal decoding, we first decompose it into components aligned with and orthogonal to the language prior subspace. The projection onto the prior subspace is given by:

$$h_{\text{proj}} = V_{\text{prior}} V_{\text{prior}}^\top h_{\text{orig}}. \tag{11}$$

While $h_{\text{proj}}$ captures prior-related components, indiscriminately removing it may cause Manifold Departure. We therefore introduce an adaptive mechanism that selectively suppresses prior components only when they conflict with visual evidence.

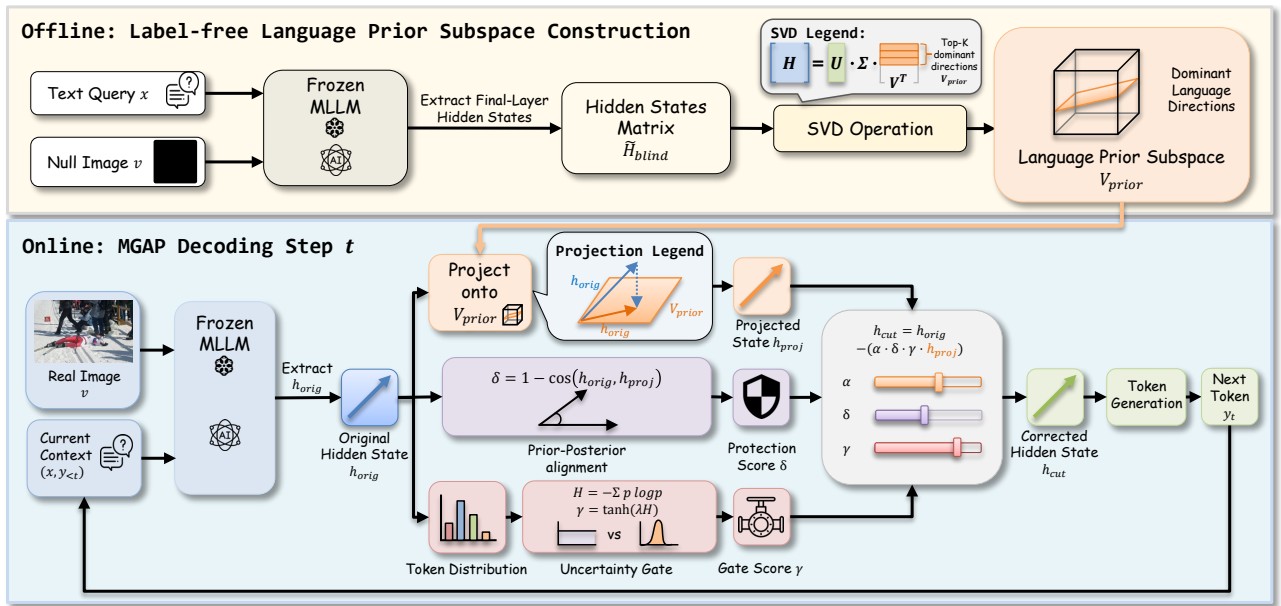

*Figure 4.* Overview of the proposed Manifold-Guided Adaptive Projection (MGAP). The language-prior subspace is constructed offline using unlabeled blind inputs, while decoding-time intervention is performed online via geometry-aware adaptive projection.

**Prior-Posterior alignment**   We quantify the agreement between the original hidden state and its prior projection using cosine similarity:

$$\delta = 1 - \cos(h_{\text{orig}}, h_{\text{proj}}). \tag{12}$$

A small $\delta$ indicates that the current state is consistent with the prior subspace and does not require additional suppression, whereas a large $\delta$ signals a mismatch that warrants stronger attenuation of prior components.

**Uncertainty-Aware gating**   Hallucinations are often accompanied by increased predictive uncertainty. We measure uncertainty using the Shannon entropy of the token distribution:

$$H = -\sum_{y} p(y) \log p(y), \tag{13}$$

and derive a gating factor:

$$\gamma = \tanh(\lambda H), \tag{14}$$

which amplifies intervention when generation is unstable and suppresses it when the model is confident.

**Adaptive projection**   The corrected hidden state is computed as:

$$h_{\text{cut}} = h_{\text{orig}} - \alpha \cdot \gamma \cdot \delta \cdot h_{\text{proj}}. \tag{15}$$

Equivalently, letting $\beta = \alpha \cdot \gamma \cdot \delta$, MGAP applies $h_{\text{cut}} = h_{\text{orig}} - \beta h_{\text{proj}}$. When visual evidence and language priors are consistent, both $\gamma$ and $\delta$ become small, reducing the operation to an approximate identity mapping. As a result, MGAP avoids the global linear translation that causes Manifold Departure in prior methods.

### 4.3. Theoretical Insights

To clarify why MGAP avoids the indiscriminate behavior of uniform linear suppression, we summarize three theoretical properties below. The proofs are provided in Appendix B.

**Theorem 4.1** (Distance decrease under prior-aligned residual). *Let $h_{\text{orig}}$ denote the original decoding hidden state, $h_{\text{proj}}$ its projection onto the learned prior subspace, and $h_{\text{cut}} := h_{\text{orig}} - \beta h_{\text{proj}}$ the MGAP-corrected state with $\beta > 0$. Under a* prior-aligned residual *condition (formalized in Appendix B), MGAP yields:*

$$\|h_{\text{cut}} - h_{\text{gt}}\|^2 < \|h_{\text{orig}} - h_{\text{gt}}\|^2. \tag{16}$$

**Interpretation**   When the current error component lies along the prior projection direction $h_{\text{proj}}$, subtracting (a suitable amount of) $h_{\text{proj}}$ moves the hidden state closer to the ground-truth-aligned state $h_{\text{gt}}$. This analysis is illustrative and does not assume access to $h_{\text{gt}}$ at inference

**Theorem 4.2** (Bounded coefficient and bounded step). *Assume $\beta := \alpha \cdot \gamma \cdot \delta$ with $\gamma := \tanh(\lambda H)$ and $\delta := 1 - \cos(h_{\text{orig}}, h_{\text{proj}})$. Then $0 \leq \beta < \alpha$, and the update magnitude is bounded:*

$$\|h_{\text{cut}} - h_{\text{orig}}\| \leq \alpha \|h_{\text{orig}}\|. \tag{17}$$

**Interpretation**   Since $\gamma$ and $\delta$ scales the step size, the effective coefficient $\beta$ is always bounded by $\alpha$, and the update magnitude is controlled. This prevents overly large corrections and yields stable decoding-time intervention.

**Theorem 4.3** (Subspace-selective update). *Let $V_{\text{prior}} \in \mathbb{R}^{d \times K}$ be the prior-subspace basis, and the prior projection*

*Table 1.* Main results on the POPE benchmark. We report Accuracy (%), Precision (%), and F1-score (%) on three splits. Our method consistently outperforms existing training-free decoding strategies across both backbones.

| Backbone | Method | Random | | | Popular | | | Adversarial | | |
|---|---|---|---|---|---|---|---|---|---|---|
| | | Acc. | Prec. | F1 | Acc. | Prec. | F1 | Acc. | Prec. | F1 |
| LLaVA v1.5 7B | Vanilla | 88.88 | 89.70 | 89.76 | 86.23 | 83.28 | 86.80 | 80.16 | 74.93 | 82.05 |
| | VCD | 87.57 | 86.23 | 87.79 | 84.23 | 80.76 | 85.08 | 78.56 | 73.47 | 80.66 |
| | ICD | 89.47 | 92.84 | 89.04 | 87.50 | 89.04 | 87.24 | 82.70 | 81.10 | 83.13 |
| | HalTrapper | 88.67 | 88.77 | 88.65 | 85.40 | 83.31 | 85.84 | 80.30 | 76.02 | 81.80 |
| | DeCo | 89.86 | 92.41 | 89.58 | 87.72 | 89.36 | 87.31 | 83.18 | 82.47 | 83.71 |
| | MoD | 89.24 | 91.37 | 89.17 | 87.03 | 87.96 | 86.91 | 82.51 | 80.74 | 83.02 |
| | Ours | **90.63** | **93.69** | **90.29** | **88.10** | **91.50** | **87.59** | **84.59** | **85.06** | **84.46** |
| Qwen3-VL 8B | Vanilla | 91.53 | 89.08 | 91.79 | 84.20 | 78.28 | 85.70 | 80.00 | 73.20 | 82.56 |
| | VCD | 90.67 | 88.80 | 90.83 | 83.27 | 79.41 | 84.53 | 81.36 | 74.92 | 83.60 |
| | ICD | 91.74 | 91.21 | 91.75 | 84.80 | 80.63 | 85.90 | 82.52 | 76.53 | 83.89 |
| | HalTrapper | 91.60 | 92.51 | 91.93 | 85.94 | 82.20 | 86.06 | 82.14 | 78.45 | 82.57 |
| | DeCo | **91.96** | 92.36 | **91.98** | 86.01 | 83.02 | 83.81 | 82.74 | 79.18 | 83.81 |
| | MoD | 91.87 | 91.94 | 91.86 | 85.76 | 82.71 | 83.74 | 82.61 | 78.92 | 83.74 |
| | Ours | 91.83 | **93.67** | 91.94 | **86.40** | **84.13** | **86.84** | **83.13** | 79.30 | **84.17** |

*Table 2.* Results on the CHAIR benchmark. Lower CHAIR scores indicate fewer hallucinated objects. Our method significantly reduces hallucination while maintaining competitive precision and F1.

| Method | LLaVA v1.5 7B | | | | Qwen3-VL 8B | | | |
|---|---|---|---|---|---|---|---|---|
| | CHAIRs↓ | CHAIRi↓ | Prec. | F1 | CHAIRs↓ | CHAIRi↓ | Prec. | F1 |
| Vanilla | 47.4 | 23.5 | 70.8 | 65.0 | 33.6 | 8.7 | 81.5 | 67.7 |
| HalTrapper | 47.2 | 23.5 | 70.6 | 64.9 | 30.8 | 8.8 | 81.6 | 62.3 |
| VCD | 52.8 | 15.8 | 72.6 | 76.5 | 38.4 | 11.7 | 79.5 | **68.8** |
| ICD | 51.8 | 14.7 | 73.7 | 77.4 | 34.2 | 8.4 | 81.4 | 68.5 |
| CODE | 49.8 | 13.8 | 76.0 | 75.7 | 31.4 | 8.7 | 81.6 | 64.0 |
| Ours | **26.2** | **7.6** | **85.9** | 77.4 | **26.8** | 8.1 | **83.9** | 66.2 |

$h_{\text{proj}} := V_{\text{prior}} V_{\text{prior}}^{\top} h_{\text{orig}}$. *MGAP only changes the prior component and preserves the orthogonal component:*

$$h_{\text{cut}} - V_{\text{prior}} V_{\text{prior}}^{\top} h_{\text{cut}} = h_{\text{orig}} - V_{\text{prior}} V_{\text{prior}}^{\top} h_{\text{orig}}. \quad (18)$$

**Interpretation** MGAP only modifies the prior-subspace component and leaves all non-prior (orthogonal) semantics unchanged, avoiding a global shift.

**Takeaway** MGAP applies a bounded, subspace-selective correction: its intervention strength is provably bounded (Theorem 4.2), while preserving the orthogonal semantics (Theorem 4.3) and avoiding a global shift. Moreover, when the current error is positively aligned with the prior projection (i.e., $\langle h_{\text{orig}} - h_{\text{gt}}, h_{\text{proj}} \rangle > 0$), subtracting (a suitable amount of) $h_{\text{proj}}$ provably moves the hidden state closer to a ground-truth-aligned reference (Theorem 4.1).

## 5. Experiments

### 5.1. Experimental Setup

We evaluate MGAP on two representative multimodal large language models, LLaVA v1.5-7B and Qwen3-VL-8B. Hallucination mitigation performance is assessed on POPE and CHAIR. For each backbone, we construct the language prior subspace offline using blind inputs (null image). We sample $N = 50$ textual prompts and collect the corresponding final-layer blind hidden states to form $\tilde{H}_{\text{blind}}$ (Eq. (8)). We then perform SVD and retain the top-$K$ right singular vectors $V_{\text{prior}}$ with $K = 5$ (Eq. (10)). All experiments are conducted on a single NVIDIA RTX A6000 GPU (48GB).

**Baselines.** We compare MGAP with several representative training-free decoding methods: VCD (Leng et al., 2024), which suppresses language priors via contrastive decoding with an unconditional branch; ICD (Wang et al., 2024c), which applies instruction-level contrastive decoding to pe-

*Table 3.* Ablation study on the POPE benchmark. "w/o Prot" removes consistency protection, and "w/o Gate" removes uncertainty gating. Removing either component leads to unstable trade-offs.

| Backbone | Method | Random | | | Popular | | | Adversarial | | |
|---|---|---|---|---|---|---|---|---|---|---|
| | | Acc. | Prec. | F1 | Acc. | Prec. | F1 | Acc. | Prec. | F1 |
| LLaVA v1.5 7B | w/o Prot | 87.70 | 97.24 | 86.32 | 86.60 | 94.64 | 77.60 | 83.00 | 88.99 | 82.91 |
| | w/o Gate | 86.57 | **98.41** | 84.69 | 85.63 | **96.04** | 83.80 | 83.73 | **91.61** | 82.03 |
| | Ours | **90.13** | 94.79 | **89.59** | **87.90** | 90.35 | **87.56** | **83.93** | 83.10 | **84.00** |
| Qwen3-VL 8B | w/o Prot | 91.63 | 90.25 | 91.82 | 85.15 | 82.39 | 86.02 | 80.95 | 76.40 | 83.52 |
| | w/o Gate | 91.70 | 91.27 | 91.53 | 85.37 | 80.86 | 86.36 | 82.03 | 75.18 | 83.01 |
| | Ours | **91.83** | **93.67** | **91.94** | **86.40** | **84.13** | **86.84** | **83.13** | **79.30** | **84.17** |

*Table 4.* Ablation results on the CHAIR benchmark. We analyze the contribution of consistency protection (*w/o Prot*) and uncertainty-aware gating (*w/o Gate*) on two representative backbones. Lower CHAIR scores indicate fewer hallucinated objects.

| Method | LLaVA v1.5 7B | | | | Qwen3-VL 8B | | | |
|---|---|---|---|---|---|---|---|---|
| | CHAIRs↓ | CHAIRi↓ | Prec. | F1 | CHAIRs↓ | CHAIRi↓ | Prec. | F1 |
| Vanilla | 47.4 | 23.5 | 70.8 | 65.0 | 33.6 | 8.7 | 81.5 | **67.7** |
| w/o Prot | 31.6 | 15.2 | 82.2 | 72.8 | 29.3 | 8.6 | 82.4 | 65.9 |
| w/o Gate | 35.4 | 8.1 | 80.2 | 71.1 | 28.8 | 8.1 | 83.3 | 66.1 |
| Ours | **26.2** | **7.6** | **85.9** | **77.4** | **26.8** | **8.0** | **83.9** | 66.2 |

nalize prior-driven generations; CODE (Kim et al., 2024), which contrasts self-generated descriptions to reduce hallucinations; and HalTrapper (Zheng et al., 2025), a recent method that attributes hallucinations to contextual accumulation in long responses and mitigates them by dynamically regulating context influence during decoding. All baselines operate without additional training and are evaluated under their recommended settings.

## 5.2. Main Results

We evaluate MGAP on the POPE benchmark using two representative backbones, LLaVA v1.5-7B and Qwen3-VL-8B. Results are summarized in Table 1.

**Main Results on POPE** Across all splits and both backbones, MGAP consistently achieves the best performance. Compared to vanilla decoding, our method improves F1 on all POPE splits, while avoiding the performance degradation observed in existing contrastive decoding approaches. On the *Random* and *Popular* splits, MGAP achieves clear precision gains, indicating more faithful object grounding. On the *Adversarial* split, which explicitly targets hallucination failure modes, MGAP yields the strongest overall performance on both backbones, demonstrating robustness under severe visual–linguistic conflict. Compared with training-free baselines such as VCD, ICD, and HalTrapper, MGAP consistently achieves superior trade-offs across all splits. In particular, VCD underperforms vanilla decoding even on non-adversarial splits, confirming that indiscriminate linear

suppression of language priors disrupts valid semantic configurations. By contrast, MGAP selectively suppresses prior components only when they conflict with visual evidence, thereby avoiding unnecessary interference. As a result, our method is the only approach that simultaneously improves precision and accuracy while maintaining or improving F1 across all evaluation settings.

**Main Results on CHAIR** We further evaluate MGAP on the CHAIR benchmark, which measures object-level hallucinations in image captioning. Results are reported in Table 2. Across both backbones, MGAP achieves a substantial reduction in hallucination rates, significantly lowering both CHAIRs and CHAIRi compared to all baselines. Notably, this reduction is accompanied by clear gains in precision, indicating that hallucination mitigation is achieved without overly conservative generation. In contrast, contrastive decoding baselines (e.g., VCD and ICD) exhibit mixed behavior across metrics and backbones, often improving only one of CHAIRs / CHAIRi while worsening the other, reflecting an unstable trade-off in object grounding. These results demonstrate that MGAP generalizes beyond binary question answering to open-ended captioning tasks.

## 5.3. Generalization to Additional Benchmarks

While POPE primarily evaluates object-level hallucinations through binary probing questions, it remains unclear whether the learned hallucination priors generalize beyond this specific evaluation protocol. To further assess the ro-

*Table 5.* Results on the AMBER benchmark using LLaVA v1.5 7B. ↓ indicates lower is better, and ↑ indicates higher is better. Best results are highlighted in **bold**.

| Model | CHAIR↓ | Cover↑ | Hal↓ | Cog↓ |
|---|---|---|---|---|
| LLaVA v1.5 7B | 11.2 | 50.2 | 47.9 | 4.6 |
| + VCD | 8.9 | 51.2 | 38.1 | 4.4 |
| + ICD | 8.6 | 51.1 | 37.3 | 3.9 |
| + CODE | 9.0 | 51.1 | 39.5 | 4.3 |
| + HalTrapper | 8.0 | 51.5 | 36.3 | **3.8** |
| + MGAP (Ours) | **7.6** | **51.7** | **35.1** | **3.8** |

*Table 6.* Inference efficiency comparison on 3000 samples. We report total inference time and average latency per sample. Our method is significantly more efficient than contrastive decoding baselines.

| Method | Total Time | Sec / Sample |
|---|---|---|
| Vanilla | 23:57 | 0.48 |
| VCD | 1:46:54 | 2.14 |
| ICD | 1:45:18 | 2.11 |
| HalTrapper | 1:50:03 | 2.20 |
| Ours | **52:24** | **1.05** |

bustness of MGAP, we additionally evaluate on the AMBER benchmark, which contains more diverse hallucination scenarios and compositional visual reasoning challenges.

Table 5 shows that MGAP consistently improves over existing decoding-time baselines across both backbones. In particular, MGAP achieves stronger hallucination mitigation while better preserving generation fidelity, suggesting that the learned prior subspace captures hallucination-inducing directions that generalize across benchmarks.

These results indicate that hallucination priors are not merely artifacts of the POPE benchmark, but instead exhibit transferable structural properties in the representation space. This supports our hypothesis that selective prior correction can serve as a more general mechanism for mitigating hallucinations in large vision-language models.

### 5.4. Representation Analysis

To better understand the behavior of MGAP, we analyze the representation changes introduced during decoding. Specifically, we measure the average projection magnitude onto the learned prior subspace for different decoding strategies.

As shown in Table 7, uniform suppression methods tend to produce larger global representation shifts, whereas MGAP introduces more localized corrections within the learned subspace. This behavior is consistent across both backbones.

These observations align with the design motivation of

MGAP. Instead of globally suppressing decoding trajectories, MGAP performs targeted representation correction using the learned prior subspace, which helps mitigate hallucinations while better preserving semantic fidelity.

### 5.5. Ablation Study

We conduct ablation studies to analyze the contribution of each component in MGAP, namely the *consistency-based protection* and the *uncertainty-aware gating*. Specifically, we consider two variants: (1) **w/o Prot**, which removes the consistency-based protection term, and (2) **w/o Gate**, which removes the uncertainty-aware gating mechanism. Results are reported on both POPE and CHAIR benchmarks.

**Ablation Results on POPE** Table 3 summarizes the ablation results on the POPE benchmark. Removing either component leads to a noticeable degradation in overall performance, confirming that both are necessary for stable hallucination mitigation. A notable phenomenon is that both w/o Prot and w/o Gate variants exhibit abnormally high precision, particularly on LLaVA v1.5-7B. However, this precision gain comes at the cost of significantly reduced F1 scores and Accuracy. This behavior indicates an imbalanced trade-off: the model becomes overly conservative, suppressing a large portion of predictions to avoid hallucinations, but failing to correctly answer valid questions. In contrast, the full MGAP achieves a more balanced trade-off. By jointly modeling visual–linguistic consistency and generation uncertainty, our method selectively suppresses prior components only when necessary. As a result, MGAP maintains high precision while substantially improving F1 score, demonstrating stable and reliable hallucination mitigation.

**Ablation Results on CHAIR** Results on CHAIR in Table 4 show a consistent trend. While removing either protection or gating reduces hallucination rates compared to vanilla decoding, both variants exhibit inferior precision and F1 compared to the full model, indicating that both components are necessary for stable and accurate grounding.

### 5.6. Efficiency Analysis

In addition to effectiveness, inference efficiency is a critical factor for training-free decoding methods. Many contrastive decoding approaches mitigate hallucinations by performing multiple forward passes with different contexts (Leng et al., 2024; Wang et al., 2024c; Zheng et al., 2025), significantly increasing inference latency. We compare the inference efficiency of MGAP with vanilla decoding and representative baselines, including VCD, ICD, and HalTrapper. As shown in Table 6, contrastive methods like VCD, ICD, and HalTrapper incur more than twice the inference time per sample compared to vanilla decoding due to multiple contrastive branches. In contrast, MGAP only applies

*Table 7.* Number of off-manifold samples (out of 3000) under different extrapolation coefficients $\alpha$. Lower values indicate that the representations are better constrained within the valid data manifold. Best results are in **bold**, and second best are underlined.

| Method | Extrapolation Coefficient $\alpha$ | | | | |
|---|---|---|---|---|---|
| | $\alpha = 0$ | $\alpha = 0.25$ | $\alpha = 0.5$ | $\alpha = 0.75$ | $\alpha = 1.0$ |
| LLaVA v1.5 7B | 150 | – | – | – | – |
| + VCD | 150 | 515 | 2589 | 2999 | 3000 |
| + ICD | 150 | 416 | 679 | 1175 | 1887 |
| + CODE | 150 | 775 | 1642 | 2403 | 2944 |
| + HalTrapper | 150 | 2098 | 2672 | 2918 | 2989 |
| + MGAP (Ours) | **150** | **76** | **210** | **285** | **402** |
| Qwen3-VL 8B | 121 | – | – | – | – |
| + VCD | 121 | 448 | 2386 | 2979 | 3000 |
| + ICD | 121 | 362 | 694 | 1187 | 1804 |
| + CODE | 121 | 724 | 1548 | 2297 | 2886 |
| + HalTrapper | 121 | 1961 | 2604 | 2881 | 2970 |
| + MGAP (Ours) | **121** | **14** | **32** | **363** | **438** |

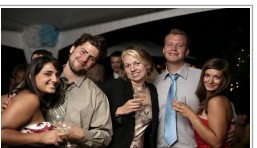

**Prompt:** Please help me describe the image in detail.

**Contrastive Decoding:** In the scene, there are two wine glasses, one held by a man and the other by a woman. Additionally, there are two bottles, one near the center of the group and the other towards the right side. The group appears to be attending a social event or gathering, as they are all dressed up and holding wine glasses.

**Ours:** The image features a group of five people standing together, posing for a picture. Three of them having wine glasses in their hands. The group consists of two men and three women, all smiling and enjoying the moment. The group is standing close to each other, creating a sense of camaraderie and friendship.

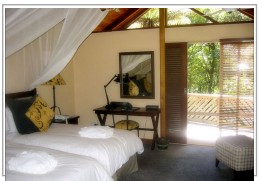

**Prompt:** Please help me describe the image in detail.

**Contrastive Decoding:** A cozy bedroom is shown with neatly made beds and soft lighting. A desk with a chair is located near the bed, providing a comfortable workspace. A table is also present in the room, accompanied by a potted plant, adding a touch of greenery to the space. The room is connected to a balcony, offering a view of the outdoors. Additional items in the room include a remote control, a cup, and a book, suggesting that the occupant may enjoy some leisure time in the bedroom.

**Ours:** The image features a cozy bedroom with a large bed situated in the center of the room. The bed is adorned with a white comforter and has a few pillows on it. A trash can is placed near the door. A chair is placed near the bed, and a desk with a mirror is located in the room as well. The room also has a doorway leading to a balcony, providing a view of the outdoors.

*Figure 5.* Qualitative comparison on the CHAIR benchmark. VCD introduces hallucinated object mentions (highlighted in red), while MGAP produces grounded descriptions without hallucinated objects.

a lightweight intervention during a single forward pass, reducing inference latency by approximately 50 % compared to contrastive baselines, while remaining fully training-free. This efficiency–effectiveness trade-off makes MGAP particularly suitable for practical deployment.

### 5.7. Qualitative Case Study

We present qualitative case studies from CHAIR to illustrate how MGAP mitigates hallucinations. Figure 5 shows representative examples comparing Visual Contrastive Decoding (VCD) and MGAP. In these cases, VCD hallucinates objects that are not supported by the visual content, such as chairs or tableware, likely due to over-reliance on language priors associated with indoor scenes. These hallucinations lead to reduced description faithfulness. In contrast, MGAP avoids introducing unsupported object mentions. By selectively regulating language priors based on visual–linguistic consistency and generation uncertainty, our method preserves informative descriptions while suppressing halluci-

nated content. Importantly, the improvement does not stem from overly conservative generation: MGAP still provides detailed and coherent captions grounded in visual evidence. These qualitative results align with the quantitative findings in Sections 5.2 and 5.5, indivating that MGAP suppresses object hallucinations while maintaining descriptive quality.

## 6. Conclusion

We identify a common failure mode of training-free decoding for hallucination mitigation: indiscriminate linear prior suppression disrupts hidden representations, which we term *Manifold Departure*. We propose Manifold-Guided Adaptive Projection (MGAP), which models a language-prior subspace and applies uncertainty-aware selective projection at decoding time.[1] Experiments prove improved trade-offs between hallucination reduction and descriptive fidelity.

---

[1]Our code and models are publicly available at: https://github.com/ZJU-OmniAI/MGAP

## Impact Statement

This paper advances multimodal language models by improving hallucination mitigation. We anticipate positive impacts in applications like visual question answering, with continued attention to fairness and robustness.

## Acknowledgements

This work is supported by the Zhejiang Pioneer Project under Grant No. 2025C06SA201986 and the Zhejiang Key Laboratory Project (2024E10001).

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

# A. Hyper-parameter Sensitivity Analysis

In this section, we investigate the sensitivity of our proposed MGAP to the hyper-parameters $K$ and $N$. The experiments are conducted on the LLaVA v1.5 7B backbone using the POPE benchmark. Table 8 reports the Accuracy, Precision, and F1 scores across the Random, Popular, and Adversarial evaluation splits under different settings of $K \in \{1, 3, 5, 10\}$ and $N \in \{10, 50, 100, 200\}$.

Remarkably, as shown in Table 8, MGAP demonstrates exceptional robustness to hyper-parameter variations. The performance metrics remain highly stable across all tested values of $K$ and $N$, indicating that our method does not rely on meticulous parameter tuning to achieve superior hallucination mitigation. Specifically, increasing $K$ from 1 to 5 brings slight improvements, particularly on the more challenging Adversarial split, while further increasing $K$ to 10 yields no additional gains. Similarly, regarding the parameter $N$, the performance reaches its peak around $N = 50$. Expanding $N$ beyond 50 does not significantly improve the generation quality but inevitably introduces additional computational overhead. Therefore, to achieve an optimal balance between anti-hallucination performance and computational efficiency, we adopt $K = 5$ and $N = 50$ as the default configuration for all main experiments.

*Table 8.* Sensitivity analysis of hyper-parameters $K$ and $N$ on the POPE benchmark using LLaVA v1.5 7B. Metrics are reported across Random (R), Popular (P), and Adversarial (A) splits in the format of R / P / A. The best result for each individual split within the $K$ and $N$ blocks is marked in **bold**.

| Parameter | Value | Acc (R/P/A) | Prec (R/P/A) | F1 (R/P/A) |
|---|---|---|---|---|
| $K$ | 1 | 90.60 / 87.87 / 82.70 | 92.89 / 87.82 / 79.64 | 89.34 / **87.87** / 84.35 |
| | 3 | 90.53 / **88.80** / 84.10 | **93.87** / **91.62** / **85.49** | 90.16 / 87.73 / 83.69 |
| | 5 (Default) | **90.63** / 88.10 / **84.59** | 93.69 / 91.50 / 85.06 | **90.29** / 87.59 / **84.46** |
| | 10 | **90.63** / 87.77 / 82.97 | 93.38 / 87.99 / 84.24 | 90.13 / 87.73 / 83.70 |
| $N$ | 10 | 90.40 / 87.93 / 83.43 | 93.10 / 89.30 / 81.68 | 89.98 / **87.72** / 83.38 |
| | 50 (Default) | **90.63** / 88.10 / **84.59** | 93.69 / **91.50** / 85.06 | **90.29** / 87.59 / **84.46** |
| | 100 | 90.37 / 88.83 / 84.20 | **94.10** / 91.16 / **85.36** | 89.94 / 87.62 / 83.68 |
| | 200 | 90.33 / **88.95** / 84.17 | 94.03 / 91.27 / 85.31 | 89.91 / 87.56 / 83.65 |

# B. Theoretical Properties of MGAP

This appendix provides several theoretical properties of MGAP that motivate its *geometry-aware* and *stable* behavior during decoding. We emphasize that MGAP performs a *low-rank* intervention confined to a learned language-prior subspace, and the overall correction magnitude is provably bounded. These results help explain why MGAP avoids the "global translation" effect that may cause manifold departure in linear suppression methods.

## B.1. Notation and Setup

Let $V \in \mathbb{R}^{d \times K}$ denote the language prior basis with orthonormal columns, i.e., $V^\top V = I$. Define the orthogonal projection onto the prior subspace as

$$P := VV^\top. \tag{19}$$

At decoding step $t$, MGAP forms the prior projection

$$h_{\text{proj}} := Ph_{\text{orig}}, \tag{20}$$

and applies the adaptive correction

$$h_{\text{cut}} := h_{\text{orig}} - \beta\, h_{\text{proj}}, \quad \text{where} \quad \beta := \alpha \cdot g \cdot p. \tag{21}$$

Here $g$ is the uncertainty gate (computed from entropy) and $p$ is the consistency-based protection factor:

$$g := \tanh(\lambda H), \qquad p := 1 - \cos(h_{\text{orig}}, h_{\text{proj}}). \tag{22}$$

For brevity, we analyze one decoding step and omit the time index.

## B.2. Property I: MGAP Moves Toward a Ground-truth-Aligned State Under Prior-Aligned Error

**Intuition**  The previous property ensures MGAP does not over-correct, but it does not explain *when* the correction is beneficial. This property states a sufficient condition under which subtracting the prior projection reduces the distance to a ground-truth-aligned reference state. Concretely, if the current error is positively aligned with the prior component, then removing (part of) that component yields a strict improvement.

**Theorem B.1** (Ground-truth-directed correction under prior-aligned error).  *Let $h_{\mathrm{gt}}$ be a reference hidden state aligned with the ground-truth continuation (e.g., obtained by teacher forcing), and define the error $e := h_{\mathrm{orig}} - h_{\mathrm{gt}}$. Let $h_{\mathrm{proj}} = Ph_{\mathrm{orig}}$ and $h_{\mathrm{cut}} = h_{\mathrm{orig}} - \beta h_{\mathrm{proj}}$ with $\beta \geq 0$. If $\langle e, h_{\mathrm{proj}} \rangle > 0$, then for any*

$$0 < \beta < \frac{2\langle e, h_{\mathrm{proj}} \rangle}{\|h_{\mathrm{proj}}\|^2},$$

*the update strictly decreases the squared distance to $h_{\mathrm{gt}}$:*

$$\|h_{\mathrm{cut}} - h_{\mathrm{gt}}\|^2 < \|h_{\mathrm{orig}} - h_{\mathrm{gt}}\|^2.$$

*Moreover, the optimal step size (minimizing $\|h_{\mathrm{cut}} - h_{\mathrm{gt}}\|^2$ over $\beta \geq 0$) is*

$$\beta^\star = \frac{\langle e, h_{\mathrm{proj}} \rangle}{\|h_{\mathrm{proj}}\|^2}.$$

*Proof.*  We expand the squared distance:

$$\|h_{\mathrm{cut}} - h_{\mathrm{gt}}\|^2 = \|h_{\mathrm{orig}} - \beta h_{\mathrm{proj}} - h_{\mathrm{gt}}\|^2 = \|e - \beta h_{\mathrm{proj}}\|^2 = \|e\|^2 - 2\beta\langle e, h_{\mathrm{proj}} \rangle + \beta^2\|h_{\mathrm{proj}}\|^2.$$

If $\langle e, h_{\mathrm{proj}} \rangle > 0$, the quadratic is strictly smaller than $\|e\|^2$ whenever

$$-2\beta\langle e, h_{\mathrm{proj}} \rangle + \beta^2\|h_{\mathrm{proj}}\|^2 < 0 \quad \Leftrightarrow \quad 0 < \beta < \frac{2\langle e, h_{\mathrm{proj}} \rangle}{\|h_{\mathrm{proj}}\|^2}.$$

The minimizer over $\beta \geq 0$ follows by setting the derivative to zero, yielding $\beta^\star = \langle e, h_{\mathrm{proj}} \rangle / \|h_{\mathrm{proj}}\|^2$.  □

## B.3. Property II: MGAP Has a Bounded Step Size

**Intuition**  Decoding-time interventions can be brittle if the correction magnitude becomes too large. MGAP prevents this by making the effective coefficient $\beta = \alpha \cdot \mathrm{Gate} \cdot \mathrm{Protection}$ automatically bounded, which in turn bounds the update step $\|h_{\mathrm{cut}} - h_{\mathrm{orig}}\|$. This yields a conservative, non-explosive intervention in a single forward pass.

**Theorem B.2** (Bounded coefficient and bounded step).  *Assume $V^\top V = I$ and $h_{\mathrm{orig}} \neq 0$. Let $P = VV^\top$, $h_{\mathrm{proj}} = Ph_{\mathrm{orig}}$, and define*

$$\mathrm{Gate} = \tanh(\lambda H), \quad \mathrm{Protection} = 1 - \cos(h_{\mathrm{orig}}, h_{\mathrm{proj}}), \quad \beta = \alpha \cdot \mathrm{Gate} \cdot \mathrm{Protection}.$$

*Then*

$$0 \leq \mathrm{Gate} < 1, \qquad 0 \leq \mathrm{Protection} \leq 1, \qquad 0 \leq \beta < \alpha.$$

*Moreover, the MGAP update $h_{\mathrm{cut}} = h_{\mathrm{orig}} - \beta h_{\mathrm{proj}}$ satisfies*

$$\|h_{\mathrm{cut}} - h_{\mathrm{orig}}\| = \beta\|h_{\mathrm{proj}}\| \leq \alpha\|h_{\mathrm{orig}}\|.$$

*Proof.*  Since $H \geq 0$ and $\lambda > 0$, we have $\tanh(\lambda H) \in [0, 1)$, hence $0 \leq \mathrm{Gate} < 1$.

Because $h_{\mathrm{proj}} = Ph_{\mathrm{orig}}$ is an orthogonal projection, we can write $h_{\mathrm{orig}} = h_{\mathrm{proj}} + (I - P)h_{\mathrm{orig}}$ with $h_{\mathrm{proj}} \perp (I - P)h_{\mathrm{orig}}$. Therefore,

$$\cos(h_{\mathrm{orig}}, h_{\mathrm{proj}}) = \frac{\langle h_{\mathrm{orig}}, h_{\mathrm{proj}} \rangle}{\|h_{\mathrm{orig}}\|\|h_{\mathrm{proj}}\|} = \frac{\|h_{\mathrm{proj}}\|^2}{\|h_{\mathrm{orig}}\|\|h_{\mathrm{proj}}\|} = \frac{\|h_{\mathrm{proj}}\|}{\|h_{\mathrm{orig}}\|} \in [0, 1],$$

which implies $\mathrm{Protection} \in [0, 1]$ and hence $0 \leq \beta < \alpha$.

Finally, $h_{\mathrm{cut}} - h_{\mathrm{orig}} = -\beta h_{\mathrm{proj}}$ so $\|h_{\mathrm{cut}} - h_{\mathrm{orig}}\| = \beta\|h_{\mathrm{proj}}\|$. Since $P$ is non-expansive, $\|h_{\mathrm{proj}}\| = \|Ph_{\mathrm{orig}}\| \leq \|h_{\mathrm{orig}}\|$, which yields $\|h_{\mathrm{cut}} - h_{\mathrm{orig}}\| \leq \alpha\|h_{\mathrm{orig}}\|$.  □

## B.4. Property III: MGAP Modifies Only the Prior Subspace

**Intuition**  A core design goal of MGAP is to avoid perturbing semantic directions that are *not* attributable to language priors. Since the correction is applied via the projection $h_{\text{proj}}$ that lies entirely in the prior subspace, the update should leave all components orthogonal to this subspace unchanged. The following theorem formalizes this subspace-selective behavior.

**Theorem B.3** (Subspace-selective update). *Let $V_{\text{prior}} \in \mathbb{R}^{d \times K}$ have orthonormal columns ($V_{\text{prior}}^\top V_{\text{prior}} = I_K$) and define $P := V_{\text{prior}} V_{\text{prior}}^\top \in \mathbb{R}^{d \times d}$. Decompose $h_{\text{orig}}$ into its subspace and orthogonal components:*

$$h_{\text{orig}} = h_\| + h_\perp, \qquad h_\| := P h_{\text{orig}}, \ \ h_\perp := (I_d - P) h_{\text{orig}},$$

*where $I_d$ is the $d \times d$ identity matrix. Then MGAP yields*

$$h_{\text{cut}} = (1 - \beta)\, h_\| + h_\perp.$$

*In particular, the orthogonal component is preserved:*

$$(I_d - P) h_{\text{cut}} = (I_d - P) h_{\text{orig}}.$$

**Equivalent form.** Since $P = V_{\text{prior}} V_{\text{prior}}^\top$, we have

$$(I_d - P) h = h - P h = h - V_{\text{prior}} V_{\text{prior}}^\top h.$$

Therefore,

$$(I_d - P) h_{\text{cut}} = (I_d - P) h_{\text{orig}}$$

is equivalent to

$$h_{\text{cut}} - V_{\text{prior}} V_{\text{prior}}^\top h_{\text{cut}} = h_{\text{orig}} - V_{\text{prior}} V_{\text{prior}}^\top h_{\text{orig}}. \tag{23}$$

*Proof.* By definition of projection, $h_{\text{proj}} = P h_{\text{orig}} = h_\|$. MGAP applies $h_{\text{cut}} = h_{\text{orig}} - \beta h_{\text{proj}}$. Substituting $h_{\text{orig}} = h_\| + h_\perp$ and $h_{\text{proj}} = h_\|$ gives

$$h_{\text{cut}} = (h_\| + h_\perp) - \beta h_\| = (1 - \beta) h_\| + h_\perp.$$

Applying $(I_d - P)$ and using $(I_d - P) h_\| = 0$ and $(I_d - P) h_\perp = h_\perp$ yields $(I_d - P) h_{\text{cut}} = h_\perp = (I_d - P) h_{\text{orig}}$. $\qquad \square$

# C. MGAP Decoding Algorithm

Algorithm 1 summarizes the decoding-time procedure of MGAP. Given an image $v$, a query $x$, and the precomputed prior basis $V_{\text{prior}}$ (Section 4.1), MGAP computes the prior projection $h_{\text{proj}}$, estimates mismatch $\delta$ and uncertainty $\gamma$, and applies the adaptive correction to obtain $h_{\text{cut}}$ for token generation.

---

**Algorithm 1** MGAP Decoding

---

**Require:**  Image $v$, query $x$, prior basis $V_{\text{prior}}$
**Ensure:**  Generated response $y = (y_1, \ldots, y_T)$
1: Initialize $y_0 \leftarrow \langle \text{BOS} \rangle$
2: **for** $t = 1$ to $T$ **do**
3:     Compute hidden state $h_{\text{orig}}$
4:     $h_{\text{proj}} \leftarrow \mathbf{V}_{\text{prior}} \mathbf{V}_{\text{prior}}^\top h_{\text{orig}}$
5:     $\delta \leftarrow 1 - \cos(h_{\text{orig}}, h_{\text{proj}})$
6:     $p(\cdot) \leftarrow \text{Softmax}(\mathcal{F}(h_{\text{orig}}))$
7:     $H \leftarrow -\sum_y p(y) \log p(y)$
8:     $\gamma \leftarrow \tanh(\lambda H)$
9:     $h_{\text{cut}} \leftarrow h_{\text{orig}} - \alpha \cdot \gamma \cdot \delta \cdot h_{\text{proj}}$
10:     Generate $y_t \sim \text{Softmax}(\mathcal{F}(h_{\text{cut}}))$
11: **end for**
12: **return** $y$

---

## D. Hyper-parameter Sensitivity

We study the sensitivity of MGAP to two key hyper-parameters: the projection scale $\alpha$ and the uncertainty-gating temperature $\lambda$ (LLaVA v1.5-7B on POPE). Overall, MGAP is stable across a broad range of settings and exhibits a clear monotonic trend on the most challenging split.

**Effect of** $\alpha$ Increasing $\alpha$ consistently improves *Precision* on all splits, and yields monotonic gains on the Adversarial split (F1: $82.05 \rightarrow 84.00$ when $\alpha$ increases from 0 to 1.0). On Random/Popular, moderate $\alpha$ (around 0.5–0.75) gives slightly higher F1, while larger $\alpha$ further boosts Precision with a small F1 trade-off, indicating a typical precision–recall shift.

*Table 9.* Sensitivity to projection scale $\alpha$ on POPE (LLaVA v1.5-7B).

| $\alpha$ | Random | | | Popular | | | Adversarial | | |
|---|---|---|---|---|---|---|---|---|---|
| | Acc | Prec | F1 | Acc | Prec | F1 | Acc | Prec | F1 |
| 0.00 | 88.88 | 89.70 | 89.76 | 86.23 | 83.28 | 86.80 | 80.16 | 74.93 | 82.05 |
| 0.25 | 90.30 | 91.21 | 90.19 | 87.17 | 85.71 | 87.42 | 81.33 | 77.07 | 82.69 |
| 0.50 | 90.43 | 92.44 | 90.20 | 87.67 | 87.37 | 87.72 | 82.13 | 78.76 | 83.12 |
| 0.75 | **90.63** | 93.69 | **90.29** | **87.90** | 88.49 | **87.81** | 83.07 | 80.58 | 83.73 |
| 1.00 | 90.13 | **94.79** | 89.59 | **87.90** | **90.35** | 87.56 | **83.93** | **83.10** | **84.00** |

**Effect of** $\lambda$ Varying $\lambda$ shows that gating is not brittle: performance remains competitive for $\lambda \in [0.25, 1.0]$. In particular, Adversarial improves substantially over $\lambda = 0$ and peaks around $\lambda = 0.5$ (Acc: $80.16 \rightarrow 84.59$; F1: $82.05 \rightarrow 84.46$), suggesting that moderate uncertainty amplification best targets conflict cases. Very large gating can slightly over-suppress on some splits (e.g., Popular F1 drops at $\lambda = 0.75$), but the overall trend remains stable.

*Table 10.* Sensitivity to gating temperature $\lambda$ on POPE (LLaVA v1.5-7B).

| $\lambda$ | Random | | | Popular | | | Adversarial | | |
|---|---|---|---|---|---|---|---|---|---|
| | Acc | Prec | F1 | Acc | Prec | F1 | Acc | Prec | F1 |
| 0.00 | 88.88 | 89.70 | 89.76 | 86.23 | 83.28 | 86.80 | 80.16 | 74.93 | 82.05 |
| 0.25 | **90.50** | 92.22 | **90.30** | 87.73 | 87.19 | **87.82** | 82.07 | 78.46 | 83.13 |
| 0.50 | 89.80 | **95.02** | 89.17 | **88.10** | **91.50** | 87.59 | **84.59** | **85.06** | **84.46** |
| 0.75 | 90.13 | 94.86 | 89.58 | 87.93 | 90.41 | 84.87 | 83.83 | 83.15 | 84.00 |
| 1.00 | 90.13 | 94.79 | 89.59 | 87.90 | 90.35 | 87.56 | 83.93 | 83.10 | 84.00 |

## E. Computational Complexity of MGAP

### E.1. Notation

Let $d$ denote the hidden dimension and $K$ the prior subspace rank with $K \ll d$. Let $N$ denote the number of blind hidden states used to construct the prior basis. Let $V \in \mathbb{R}^{d \times K}$ be the prior basis with orthonormal columns,

$$V^\top V = I_K. \tag{24}$$

Define the orthogonal projector

$$P := VV^\top \in \mathbb{R}^{d \times d}. \tag{25}$$

At each decoding step, MGAP computes

$$h_{\text{proj}} = Ph = VV^\top h, \tag{26}$$

and performs the update

$$h_{\text{cut}} = h - \beta h_{\text{proj}}, \tag{27}$$

where

$$\beta = \alpha \cdot g \cdot p. \tag{28}$$

Here $g = \tanh(\lambda H)$ and $p = 1 - \cos(h, h_{\text{proj}})$ are scalar factors.

### E.2. Online (Per-token) Complexity

**Proposition E.1** (MGAP per-step time complexity)**.** *Given $V \in \mathbb{R}^{d \times K}$, the additional time complexity incurred by MGAP at a decoding step is*

$$T_{MGAP}(d, K) = \mathcal{O}(dK). \tag{29}$$

*Proof.* Compute $u = V^\top h \in \mathbb{R}^K$:

$$u = V^\top h. \tag{30}$$

This is a matrix–vector multiply of size $K \times d$, costing

$$\mathcal{O}(dK). \tag{31}$$

Compute the projection $h_{\mathrm{proj}} = Vu \in \mathbb{R}^d$:

$$h_{\mathrm{proj}} = Vu. \tag{32}$$

This is a matrix–vector multiply of size $d \times K$, costing

$$\mathcal{O}(dK). \tag{33}$$

Compute cosine similarity requires an inner product and two norms:

$$\langle h, h_{\mathrm{proj}} \rangle, \quad \|h\|_2, \quad \|h_{\mathrm{proj}}\|_2, \tag{34}$$

which costs

$$\mathcal{O}(d). \tag{35}$$

Compute the gated scaling $\beta = \alpha gp$ is scalar-only:

$$\beta = \alpha gp, \tag{36}$$

which costs

$$\mathcal{O}(1). \tag{37}$$

Finally compute the residual update:

$$h_{\mathrm{cut}} = h - \beta h_{\mathrm{proj}}, \tag{38}$$

which costs

$$\mathcal{O}(d). \tag{39}$$

Thus the dominant cost is two $d \times K$ / $K \times d$ matrix–vector multiplies, yielding

$$T_{\mathrm{MGAP}}(d, K) = \mathcal{O}(dK). \tag{40}$$

$\square$

**Proposition E.2** (MGAP additional memory)**.** *The additional memory required by MGAP (beyond the base model) is*

$$M_{MGAP}(d, K) = \mathcal{O}(dK), \tag{41}$$

*plus $\mathcal{O}(d)$ temporary vectors per decoding step.*

*Proof.* MGAP stores the prior basis $V \in \mathbb{R}^{d \times K}$, which takes

$$\mathcal{O}(dK) \tag{42}$$

floating-point numbers. At runtime, MGAP may allocate temporary vectors $u \in \mathbb{R}^K$ and $h_{\mathrm{proj}} \in \mathbb{R}^d$, costing

$$\mathcal{O}(K) + \mathcal{O}(d) = \mathcal{O}(d). \tag{43}$$

Therefore the asymptotic additional storage is dominated by $V$:

$$M_{\mathrm{MGAP}}(d, K) = \mathcal{O}(dK). \tag{44}$$

$\square$

# F. Offline Subspace Construction Complexity

Let $H_{\text{blind}} \in \mathbb{R}^{N \times d}$ be the centered blind-state matrix:

$$H_{\text{blind}} = \begin{bmatrix} (h_{\text{blind}}^{(1)} - \bar{h})^\top \\ \vdots \\ (h_{\text{blind}}^{(N)} - \bar{h})^\top \end{bmatrix}. \tag{45}$$

MGAP computes the top-$K$ right singular vectors:

$$H_{\text{blind}} = U\Sigma V^\top, \tag{46}$$

and retains

$$V_{\text{prior}} := V_{[:,1:K]} \in \mathbb{R}^{d \times K}. \tag{47}$$

**Full SVD (exact).**

**Proposition F.1** (Offline exact SVD complexity). *Computing an exact SVD of $H_{\text{blind}} \in \mathbb{R}^{N \times d}$ has time complexity*

$$T_{\text{SVD-full}}(N,d) = \mathcal{O}\big(\min\{Nd^2,\, N^2 d\}\big), \tag{48}$$

*and memory complexity*

$$M_{\text{SVD-full}}(N,d) = \mathcal{O}(Nd). \tag{49}$$

*Proof.* Standard dense SVD for an $N \times d$ matrix has the classical bound

$$\mathcal{O}\big(\min\{Nd^2,\, N^2 d\}\big). \tag{50}$$

Storing the data matrix requires $Nd$ entries:

$$\mathcal{O}(Nd). \tag{51}$$

$\square$

**Truncated SVD (iterative, exact up to tolerance).** In practice, MGAP only needs the top-$K$ right singular vectors. This can be obtained via Krylov subspace methods (e.g., Lanczos) using repeated matrix–vector products.

**Proposition F.2** (Offline truncated SVD complexity). *Assume $K \ll \min\{N,d\}$ and the top-$K$ singular vectors are computed by an iterative method requiring $T$ iterations. Then the time complexity is*

$$T_{\text{SVD-trunc}}(N,d,K) = \mathcal{O}(T \cdot Nd), \tag{52}$$

*and the additional memory beyond storing $H_{\text{blind}}$ is*

$$M_{\text{SVD-trunc}}(N,d,K) = \mathcal{O}((N+d)K). \tag{53}$$

*Proof.* Each iteration of Krylov/Lanczos methods uses one or a constant number of multiplications by $H_{\text{blind}}$ and $H_{\text{blind}}^\top$. A matrix–vector multiply costs

$$\mathcal{O}(Nd). \tag{54}$$

With $T$ iterations, the total is

$$\mathcal{O}(TNd). \tag{55}$$

The method stores $\mathcal{O}(K)$ basis vectors in $\mathbb{R}^N$ and/or $\mathbb{R}^d$, giving

$$\mathcal{O}(NK) + \mathcal{O}(dK) = \mathcal{O}((N+d)K). \tag{56}$$

$\square$

**Randomized SVD (approximate).** A common alternative is randomized SVD, which forms a sketch of size $K + p$ with oversampling $p$. Let $q$ denote the number of power iterations.

**Proposition F.3** (Offline randomized SVD complexity). *Randomized SVD for extracting $K$ components from $H_{\text{blind}} \in \mathbb{R}^{N \times d}$ has time complexity*

$$T_{\text{SVD-rand}}(N, d, K) = \mathcal{O}(Nd(K + p) + q \cdot Nd(K + p)),\tag{57}$$

*and memory complexity*

$$M_{\text{SVD-rand}}(N, d, K) = \mathcal{O}((N + d)(K + p)).\tag{58}$$

*Proof.* Randomized SVD forms a sketch using multiplications by $H_{\text{blind}}$ (and optionally power iterations). Each pass costs $\mathcal{O}(Nd(K + p))$ for multiplying by a $(d \times (K + p))$ test matrix. With $q$ power iterations, the number of passes scales by $(1 + q)$, yielding

$$\mathcal{O}((1 + q) \cdot Nd(K + p)).\tag{59}$$

The memory stores the sketch bases in $\mathbb{R}^N$ and $\mathbb{R}^d$:

$$\mathcal{O}((N + d)(K + p)).\tag{60}$$

$\square$

**Comparison to Multi-branch Contrastive Decoding** MGAP performs a single-pass representation update and adds only $\mathcal{O}(dK)$ arithmetic per token. In contrast, contrastive decoding baselines that require $B$ forward passes per token incur a multiplicative overhead in the base model cost. Let $C_{\text{model}}$ denote the per-token compute cost of the backbone model. Then their total per-token cost scales as

$$T_{\text{contrastive}} = \Theta(B \cdot C_{\text{model}}),\tag{61}$$

whereas MGAP scales as

$$T_{\text{MGAP-total}} = \Theta(C_{\text{model}}) + \mathcal{O}(dK).\tag{62}$$

When $K \ll d$ and the backbone dominates compute, $\mathcal{O}(dK)$ is negligible relative to $C_{\text{model}}$.

