# OpenReview forum: "Mitigating Manifold Departure: Uncertainty-Aware Subspace Rectification for Trustworthy MLLM Decoding"
_ICML.cc/2026/Conference — ICML 2026 regular_

### Official Review · Reviewer_2Y7r · 2026-03-06

**Soundness:** 4
**Presentation:** 3
**Significance:** 3
**Originality:** 4
**Overall Recommendation:** 3
**Confidence:** 4

**Summary:**

This paper hypothesized that existing contrastive decoding methods can result in deviating from semantic manifold space. The proposed method includes constructing language prior subspace by performing PCA with collected logit distributions, and projecting the original hidden state. The projected hidden space is contrasted with the original logit at test time, and the interpolation ratio is determined based on uncertainty (Gate) and similarity between prior and posterior logits (Prot). The experiments include LLaVA-1.5/Qwen3-VL with POPE/CHAIR, ablation the Prot and Gate, and latency analysis.

**Compliance With Llm Reviewing Policy:**

Affirmed.

**Final Justification:**

I appreciate the authors’ rebuttal, and I believe that most of my other concerns have been addressed. That said, one remaining issue is still central to my evaluation. The main motivation of the paper is based on the reported failure of LLaVA on POPE, which, if well supported, would call into question the conclusions of prior work. However, after checking the code, I found that the APC parameter $\beta$ is set to 0.5. This seems unusually high for the APC threshold and may overly restrict the next-token candidate pool. For reference, prior contrastive decoding methods such as VCD and ConVis use $\beta = 0.1$, where $\beta$ lies in the range $[0,1]$. In a few runs on my side, I also observed similar failures under this configuration. This makes it difficult for me to conclude that the reported failure of LLaVA-1.5 on POPE reflects a fundamental limitation of VCD or an over-reliance on language priors; instead, it appears more likely to be caused by hyperparameter choice. While this setting may work in some environments or for some methods, it does not seem to transfer reliably across contrastive decoding approaches. For that reason, if $\beta=0.5$ is indeed the setting used in the paper’s experiments, I would still need to maintain my negative assessment despite the authors having addressed most of my other concerns.

**Key Questions For Authors:**

1. The results on Figure 2 are distinct from the report of VCD literature. The following-up papers related to VCD consistently report that VCD outperforms multinomial sampling and greedy search. Could the authors provide more detailed test configurations for Figure 2 to explain this discrepancy?
2. It is true that contrastive decoding based methods can undesirably reward the logit values of specific indices therefore give unreliable responses. However, adaptive plausibility constraint (APC) is introduced to penalize the false positives and have been settled as standard regularizer. Can APC change the behavior of manifold departure under linear suppression?
3. In line 211, it is mentioned that “the decoder operates in a poorly supported regime, often leading to less stable token distributions and degraded generation quality”. How can the decoder malfunction while the Eq.7 is operated after the decoder operations? It sounds like modifying the output can harm the generation of the output.
4. In the literature of contrastive decoding, we select the amateur distribution with the logits obtained by corrupted with heuristics (e.g., noise addition for VCD, language substitution for CODE) which is likely to hallucinate. Since $h_{proj}$ is a smoothed version of the original logit distribution by principle components, how can it induce weaker perception/reasoning?
5. Can the principal component matrix $V_{prior}$ generalized across different datasets and tasks, or does it require dataset-specific re-calibration?

The most significant concerns are listed in the Questions section. I truly enjoyed reading the paper and am willing to raise my score once these concerns are adequately addressed.

**Limitations:**

1. The practical severity of a hidden state not being strictly placed on the manifold is not entirely convincing. When utilizing standard search algorithms (e.g., greedy search or top-k sampling), the absolute placement of the representation on the manifold is arguably less important than the relative ranking of the top logit values. The paper needs to better justify why off-manifoldness inherently degrades performance.
2. While adaptively projecting and subtracting the hidden state from the original distribution is interesting, current script lacks explanations of which logit values are incentivized or penalized.
3. There are some further research questions that have not been addressed, including: (1) sensitivity of k (2) efficacy on general reasoning (e.g., MMBench, VQA-v2), (3) effect on models with different scales. Also, the methods compared against in the experiments were published 2 years ago (CODE, ICD), and the number of tested models is considered insufficient.

**Strengths And Weaknesses:**

Strength
1. The paper formally introduces and rigorously defines the concept of "manifold departure" with solid mathematical reasoning. This provides a fresh perspective that could open a new avenue of research in decoding strategies.
2. The proposed mechanism of projecting the hidden state onto a low-dimensional space and applying interpolation to intentionally preserve the language prior is highly creative and intuitively sound.

Weaknesses are described in the Limitation section.

---

> ### Author Rebuttal · Authors · 2026-03-28
>
> We sincerely thank the reviewer for the constructive questions.
>
> ### Q1. Why does Figure 2 show degradation for VCD, while prior literature sometimes reports gains?
>
> **We do not claim that VCD is universally harmful.** Rather, its effect depends on whether language priors are aligned with visual evidence. When prior and visual evidence are aligned, priors can be helpful and stabilizing; when they conflict, suppressing prior influence may reduce hallucination, but blunt suppression can also damage content quality and semantic consistency. Therefore, it is fully consistent with our motivation that VCD may improve some tasks while degrading others.
>
> **Other recent work has likewise reported negative effects of VCD on certain tasks (see Figure 2 in [1])**. Therefore, the degradation shown in our Figure 2 should not be interpreted as contradicting prior literature, but rather as further evidence that blunt contrastive prior suppression can help in some settings and hurt in others.
>
> For clarity, Figure 2 uses the same evaluation pipeline as our main POPE experiments on LLaVA-v1.5-7B, with the standard POPE protocol and the recommended VCD settings.
>
> ### Q2. Can APC change the behavior of manifold departure under linear suppression?
>
> Before addressing the specific questions, **we would like to clarify one important distinction already formalized in Eq. (1): decoding proceeds as “hidden state → decoder head F(⋅) → logits → softmax → probability”**. This distinction is central to our method, because MGAP intervenes at the representation level before logits are formed, whereas many contrastive variants ultimately operate through logit-level combination or correction.
>
> Therefore, APC cannot directly repair or reverse a manifold departure that has already occurred in the hidden representation space，because it penalizes the false positives in **logit-level**.
>
> As a supporting side observation, [2] argue that much of the apparent improvement of contrastive decoding on POPE may actually come from APC, rather than from the contrastive mechanism itself. This further motivates us to investigate deeper representation-level phenomena—such as hidden-state correction and manifold departure—rather than relying only on output-level regularization.
>
> ### Q3. Clarification of line 211 / Eq. (7)
> We sincerely thank the reviewer for catching this issue, and we apologize for the confusion caused by our presentation.
>
> The reviewer is correct: Eq. (7) in the current draft is mistakenly written in terms of hidden states $h$, whereas **the intended VCD-style formulation there should be written at the logits level**. This mismatch may have blurred the distinction between hidden-state and logit-level operations. This is a writing issue on our side, and we will revise it carefully.
>
> ### Q4. h_proj is not just a smoothed version
> $h_{proj}$ is not a smoothed distribution or an auxiliary contrastive branch; **it is the projection of the current hidden representation $h_{orig}$ onto the learned prior subspace $V_{prior}$**. In other words, MGAP geometrically decomposes the current hidden state into a prior-subspace component and its orthogonal component.
>
> This is also why MGAP does not weaken grounded perception or reasoning. As formalized in Theorem 4.3, MGAP modifies only the projected prior component, while preserving the orthogonal component exactly. Therefore, the corrected state suppresses only the prior-related part of the representation, without damaging the grounded multimodal/posterior part.
>
> ### Q5. Can the principal component matrix generalized across different datasets and tasks?
> The learned principal-component subspace can generalize at **the domain level**.
>
> For example, both CHAIR and AMBER are visually grounded image-generation tasks, and in our additional experiments **we directly use the CHAIR-learned subspace on AMBER without dataset-specific re-calibration**. MGAP still remains state-of-the-art in Table 1, which provides direct evidence that the learned subspace can generalize within the same domain.
> If tasks differ substantially, however, re-calibration would likely be needed.
>
> **Table 1.Results on AMBER (LLaVA v1.5 7B).**
> | AMBER | Model            | CHAIR ↓ | Cover ↑ | Hal ↓ | Cog ↓ |
> |------|------------------|--------|--------|------|------|
> |      | + LLaVa v1.5 7B    | 11.2   | 50.2   | 47.9 | 4.6  |
> |      | + VCD              | 8.9    | 51.2   | 38.1 | 4.4  |
> |      | + ICD              | 8.6    | 51.1   | 37.3 | 3.9  |
> |      | + CODE             | 9.0    | 51.1   | 39.5 | 4.3  |
> |      | + Haltrapper       | 8.0    | 51.5   | 36.3 | 3.8  |
> |      | **+ Ours**         | **7.6**| **51.7**| **35.1** | **3.8** |
>
> [1] Yin et al. ClearSight: Visual Signal Enhancement for Object Hallucination Mitigation in Multimodal Large Language Models. CVPR, 2025.
>
> [2] Yin et al. The Mirage of Performance Gains: Why Contrastive Decoding Fails to Mitigate Object Hallucinations in MLLMs. NeurIPS, 2025.

---

> > ### Author Rebuttal · Reviewer_2Y7r · 2026-04-03
> >
> > I would like to thank the authors for their detailed rebuttal, especially on additional recalibration experimental results. The followings describe my thoughts after authors' rebuttal and another careful reading of the manuscript.
> >
> > 1. Figure 2 shows the results of LLaVA-1.5-7B results on POPE benchmark comparing multinomial decoding and VCD, and I understood that VCD can harm the result if there are anomalies in the given QA. To my best understanding, authors refer to visual evidence and language prior is misaligned (eg, give an image of blue banana and asking its color as shown in Figure 1). However, **VCD paper and most of its following works include the result of LLaVA-1.5 with POPE, consistently demonstrating its superior performance over multinomial decoding** [1,2,3]. I also manually implemented the experiments by myself, and the result does not align with the author's report. Can authors share further details of this observation? There might be a bug in the implementation.
> > 2. Authors mentioned in rebuttal: "MGAP intervenes at the representation level before logits are formed, whereas many contrastive variants ultimately operate through logit-level combination or correction". But in Equation 1, **$h_t$ denotes the final-layer hidden state, and the $h_{orig}$ is projected to $V_{prior}$ as mentioned in Equation 11 and Figure 4**. I believe MGAP also intervenes the representation on next-token logit space, and principal components must be affected by excluding implausible tokens by explicitly assigning -inf to them. Please correct me if I am wrong.
> > 3. Thanks for the acknowledgement. **Please let all reviewers know how authors are planning to revise the logic**.
> > 4. Authors clarified that h_proj suppresses only the prior-related part of the representation, without damaging the grounded multimodal/posterior part rather than a simple smoothed version of h_orig. **This raises a more critical higher concern**, the purpose of contrastive decoding is to mitigate the hallucination of expert models that amateur models can make. If h_proj suppress the language prior, it may be more faithful to the given multimodal query, therefore amateur logit may have stronger representation than expert logits. **Therefore, the contrasted logit will eventually mitigate the model output to generate faithful responses**.
> > 5. Thanks for the additional experiments. I believe that experimental results inferenced with PCA subspace from different domains must be included in the manuscript to evaluate how well the method generalizes across domains. For example, subspace calibrated with POPE and inference with MathVista.
> >
> > Furthermore, I look forward to reading the authors' responses to the remaining unaddressed questions in the Limitation section.
> >
> > References
> >
> > [1] Kim et al. Code: Contrasting self-generated description to combat hallucination in large multi-modal models. NeurIPS 2024
> >
> > [2] Park et al. ConVis: Contrastive Decoding with Hallucination Visualization for Mitigating Hallucinations in Multimodal Large Language Models. AAAI 2025
> >
> > [3] Im et al. Self-Augmented Visual Contrastive Decoding. ICLR 2026

---

> > > ### Author Response · Authors · 2026-04-05
> > >
> > > We appreciate your responsible assessment. Due to the length limit, our responses here are brief; **full responses and tables are in the anonymous repository:** https://anonymous.4open.science/r/CD-baseline.
> > > ### Q1
> > > We checked the code and found no bug. **Also, such negative effects are not unique to our paper: recent work also reports degradation or unstable hallucination mitigation for CD in some settings[1,2]**. Our core code is included in the anonymous repository.
> > > ### Q2
> > > We agree that MGAP ultimately affects next-token logits. So the key issue is whether the principal components $V_{prior}$ could be corrupted by low-scoring tokens without APC.
> > >
> > > In our method, $V_{prior}$ uses **only the top-K components** of $H_{blind}$ (Eq. 9–10). This **naturally suppresses low-variance directions—including noisy or implausible-token components—without token-level pruning**.
> > >
> > > K-sensitivity analysis (Table R3 in repo) confirms that small K captures main prior directions while excluding noise, whereas larger K gradually introduces noisy components. Thus, robustness comes from the top-K construction, not APC.
> > > ### Q3
> > > First, we will distinguish three levels: MGAP acts on hidden states; prior contrastive baselines operate on logits or via branch combination; both ultimately affect next-token logits.
> > >
> > > Second, we will revise line 211 to avoid suggesting decoder “malfunction.” Our point is that **uniform suppression can push the hidden state too far from normal decoding trajectories into a weakly supported region that the decoder is less familiar with**, yielding less stable output distributions.
> > > ### Q4
> > > This concern may stem from our shorthand that “MGAP modifies only the projected prior component.” It has two intended meanings: (i) MGAP is restricted to the projected prior component while preserving the orthogonal component, (ii) MGAP is **applied selectively rather than unconditionally**.
> > >
> > > Our point is not that reducing prior influence is always wrong, nor that CD never helps. **The key distinction is whether suppression is indiscriminate or selective:** MGAP gates correction by prior-posterior mismatch and uncertainty (Eq. 12, 14), whereas contrastive decoding typically uses a much coarser branch-level subtractive correction.
> > >
> > > ### Q5
> > > We evaluated POPE→MathVista and MathVista→MathVista. The former shows the prior subspace is domain-dependent; the latter recovers this mismatch but brings only limited gains, likely because MathVista depends more on multi-step reasoning than prior-driven decoding errors.
> > >
> > > **Table x1. Result on MathVista**
> > > | SubSpace   | LLaVA Acc | Qwen Acc |
> > > |------------|-----------|----------|
> > > | Vanilla    | 25.0      | 76.9     |
> > > | POPE       | 22.5      | 75.6     |
> > > | MathVista  | **25.2**     | **77.0**     |
> > >
> > > **Table x2. Number of off-manifold samples under different α. Off-manifold are defined as those with kNN distance $d_k > \tau$
> > >  (as in Figure 3).**
> > > | Backbone | Method | α = 0 | α = 0.25 | α = 0.5 | α = 0.75 | α = 1.0 |
> > > |---|---|---:|---:|---:|---:|---:|
> > > | LLaVA |
> > > |  | VCD | 150 | 515 | 2589 | 2999 | 3000 |
> > > |  | ICD | 150 | 416 | 679 | 1175 | 1887 |
> > > |  | CODE | 150 | 775 | 1642 | 2403 | 2944 |
> > > |  | HalTrapper | 150 | 2098 | 2672 | 2918 | 2989 |
> > > |  | MGAP | 150 | 76 | 210 | 285 | 402 |
> > > | Qwen|
> > > |  | VCD | 121 | 448 | 2386 | 2979 | 3000 |
> > > |  | ICD | 121 | 362 | 694 | 1187 | 1804 |
> > > |  | CODE | 121 | 724 | 1548 | 2297 | 2886 |
> > > |  | HalTrapper | 121 | 1961 | 2604 | 2881 | 2970 |
> > > |  | MGAP | 121 | 143 | 236 | 321 | 438 |
> > >
> > > **Table x3. Extended baselines**
> > > | Setting | Value | Acc (R/P/A) | Prec (R/P/A) | F1 (R/P/A) |
> > > |---|---|---|---|---|
> > > | LLaVA
> > > |  | DeCo | *89.86* / *87.72* / *83.18* | 92.41 / *89.36* / *82.47* | 89.58 / *87.31* / *83.71* |
> > > |  | MoD | 89.24 / 87.03 / 82.51 | 91.37 / 87.96 / 80.74 | 89.17 / 86.91 / 83.02 |
> > > |  | Ours | **90.63** / **88.10** / **84.59** | **93.69** / **91.50** / **85.06** | **90.29** / **87.59** / **84.46** |
> > > | Qwen
> > > |  | DeCo | **91.96** / *86.01* / *82.74* | 92.36 / *83.02* / *79.18* | **91.98** / *86.29* / 83.81 |
> > > |  | MoD | 91.87 / 85.76 / 82.61 | 91.94 / 82.71 / 78.92 | 91.86 / 86.12 / 83.74 |
> > > |  | Ours | *91.83* / **86.40** / **83.13** | **93.67** / **84.13** / **79.30** | *91.94* / **86.84** / **84.17** |
> > >
> > > ### L1
> > > We additionally counted off-manifold samples on POPE and found that methods with higher off-manifold ratios perform worse, consistent with prior geometric analyses [3], which suggest that off-manifold perturbations are less reliable.
> > > ### L2
> > > MGAP does not reward fixed logits, but selectively down-weights prior-dominated ones while preserving grounded ones.
> > > ### L3
> > > We added K-sensitivity (Table R3 in the repository), newer baselines DeCo,MoD [4,5] (Table x3), and a general-reasoning benchmark (Table x1); scale study is left to revision.
> > >
> > > [1]ClearSight. CVPR 2025.
> > > [2]The Mirage of Performance Gains. NeurIPS 2025.
> > > [3]Disentangling Adversarial Robustness and Generalization. CVPR 2019.
> > > [4]Dynamic Correction Decoding. ICLR 2025
> > > [5]Mixture of Decoding. ACL 2025

---

### Official Review · Reviewer_vvTy · 2026-03-12

**Soundness:** 3
**Presentation:** 3
**Significance:** 3
**Originality:** 3
**Overall Recommendation:** 4
**Confidence:** 2

**Summary:**

This paper addresses object hallucination in Multimodal Large Language Models, where generated content contradicts visual evidence due to over-reliance on language priors. The paper proposes Manifold-Guided Adaptive Projection, a geometry-aware decoding framework. Empirically, it outperforms contrastive decoding baselines on POPE and CHAIR benchmarks across LLaVA v1.5-7B and Qwen3-VL-8B, achieving trade-offs between hallucination suppression and descriptive fidelity.

**Compliance With Llm Reviewing Policy:**

Affirmed.

**Key Questions For Authors:**

1. The paper set the num of top-K singular vectors K=5  without ablation studies. How sensitive is MGAP's performance to the choice of K ?
2. The paper construct the prior subspace using N=50  textual prompts. Is this sufficient to capture the diversity of language priors in large-scale pre-training?

**Limitations:**

Yes

**Strengths And Weaknesses:**

The submission is technically sound and well-presented, offering a clearly structured narrative that effectively identifies the geometric failure mode of Manifold Departure and proposes MGAP as a theoretically grounded, training-free solution with rigorous proofs and comprehensive experiments on standard benchmarks. The work addresses an important problem in MLLMs, providing practical utility through reduced inference latency and improved hallucination metrics while introducing novel geometric insights via subspace-selective intervention.

---

> ### Author Rebuttal · Authors · 2026-03-28
>
> We sincerely thank the reviewer for the positive assessment and the constructive questions on prior-subspace construction.
> ### Q1. Sensitivity to the choice of K
>
> MGAP is robust to moderate choices of K and N, and our default choice  lies in a stable high-performance region.
>
> In particular, K=1 tends to underfit the dominant prior directions, while K=10 slightly weakens selectivity and causes mild degradation on some splits. These results support our default choice  as a robust middle ground rather than a narrowly tuned value.
>
>
> **Table 1.Sensitivity analysis on POPE (LLaVA v1.5 7B). R/P/A = Random / Popular / Adversarial.**
> | Setting | Value | Acc (R/P/A) | Prec (R/P/A) | F1 (R/P/A) |
> |---|---:|---|---|---|
> | K | 1   | 90.60 / 87.87 / 82.70 | 92.89 / 87.82 / 79.64 | 89.34 / **87.87** / 84.35 |
> | K | 3   | 90.53 / **88.80** / 84.10 | **93.87** / **91.62** / **85.49** | 90.16 / 87.73 / 83.69 |
> | K | 5   | **90.63** / 88.10 / **84.59** | 93.69 / 91.50 / 85.06 | **90.29** / 87.59 / **84.46** |
> | K | 10  | **90.63** / 87.77 / 82.97 | 93.38 / 87.99 / 84.24 | 90.13 / 87.73 / 83.70 |
> | Setting | Value | Acc (R/P/A) | Prec (R/P/A) | F1 (R/P/A) |
> | N | 10  | 90.40 / 87.93 / 83.43 | 93.10 / 89.30 / 81.68 | 89.98 / **87.72** / 83.38 |
> | N | 50  | **90.63** / 88.10 / **84.59** | 93.69 / **91.50** / 85.06 | **90.29** / 87.59 / **84.46** |
> | N | 100 | 90.37 / 88.83 / 84.20 | **94.10** / 91.16 / **85.36** | 89.94 / 87.62 / 83.68 |
> | N | 200 | 90.33 / **88.95** / 84.17 | 94.03 / 91.27 / 85.31 | 89.91 / 87.56 / 83.65 |
>
>
> ### Q2. Is N=50 sufficient to capture language priors?
>
> Yes for POPE. On POPE, the queries are **highly templated object-probing questions (“Is there a [object] in the image?”)**, so the linguistic variation is relatively low-dimensional and a moderate prompt pool is already sufficient to capture the dominant blind-prior directions. This explains why performance quickly saturates instead of continuing to improve with substantially larger N in Table 1.
>
> Rather, **the required prompt-pool size should depend on the linguistic diversity of the target domain**.  We believe the required prompt-pool size should scale with the linguistic diversity of the target domain. If one aims to capture richer and more diverse language priors—especially those closer to the heterogeneity of large-scale pretraining corpora—a larger N would naturally be needed

---

> > ### Author Rebuttal · Reviewer_vvTy · 2026-04-03
> >
> > Thanks for the further clarification. I have no further questions.

---

> > > ### Author Response · Authors · 2026-04-06
> > >
> > > Thank you again for your positive feedback and for the time you've dedicated to reviewing our work. We are happy that our responses met your expectations.

---

### Official Review · Reviewer_E9hf · 2026-03-13

**Soundness:** 2
**Presentation:** 3
**Significance:** 2
**Originality:** 2
**Overall Recommendation:** 4
**Confidence:** 3

**Summary:**

This paper studies object hallucination in multimodal large language models (MLLMs). The authors argue that many training-free decoding methods reduce hallucination by suppressing language priors in a globally uniform way, but that this can also harm cases where the language prior is actually consistent with the image. They describe this failure mode as **manifold departure**: a decoding intervention moves hidden states away from the high-density region associated with normal decoding trajectories. To address this, the paper proposes **Manifold-Guided Adaptive Projection (MGAP)**. MGAP first builds a low-rank language-prior subspace from blind hidden states collected from null-image inputs using SVD. During decoding, it projects the current hidden state onto this subspace and attenuates only the projected component, with the intervention strength gated by prior-posterior mismatch and token-level uncertainty. Experiments on POPE and CHAIR with LLaVA v1.5-7B and Qwen3-VL-8B show improved hallucination metrics over the baselines included in the paper, while using only a single forward branch and therefore running faster than multi-branch contrastive decoding methods.

**Compliance With Llm Reviewing Policy:**

Affirmed.

**Final Justification:**

All my concerns have been solved. I would like to raise the score.

**Key Questions For Authors:**

1.  **How does MGAP compare conceptually and empirically to the closest related inference-time methods, especially Activation Steering Decoding, DeCo, and MoD?**\
    If the authors can clearly differentiate MGAP from these methods, or provide evidence that MGAP remains competitive against them, my assessment of originality and significance would improve.

2.  **Can the authors provide stronger evidence that manifold departure is a general mechanism rather than a post-hoc interpretation of one baseline?**\
    For example, does the same off-manifold trend appear consistently for ICD, CODE, or HalTrapper across both backbones and across multiple tasks? A positive answer would strengthen my assessment of the paper's main scientific claim.

3.  **How sensitive is MGAP to the construction of the prior subspace?**\
    Please discuss the effect of the blind prompt pool, `N`, `K`, layer choice, and null-image construction. If the method is robust to these choices, that would improve my view of soundness and practical value.

4.  **Can the authors add at least one broader benchmark suite beyond POPE and CHAIR, or explain why those two are sufficient for the paper's current claims?**\
    A stronger empirical scope would materially affect my assessment of both soundness and significance.

**Strengths And Weaknesses:**

### Strengths

1.  **The core intuition is reasonable and practically relevant.** The paper makes a useful point that language priors in MLLMs are not uniformly harmful. In visually aligned cases they can stabilize decoding, so globally subtracting them is indeed a blunt intervention. This is a better motivation than the simpler narrative that hallucination is always caused by excessive prior reliance.

2.  **The proposed intervention is simple and deployment-friendly.** MGAP is easy to describe, requires no retraining, and only adds a lightweight projection-and-gating step at inference time. Compared with multi-branch contrastive decoding, this design is attractive from a systems perspective.

3.  **The paper is generally readable.** The method section is clearly structured, the equations are easy to follow, and the appendices provide algorithmic and complexity details that help readers understand what is actually being implemented.

### Weaknesses

1.  **The strongest mechanism claim is not validated strongly enough.** The paper presents manifold departure as the key explanation for why prior-suppression baselines fail, but the evidence is still limited to a PCA visualization and a kNN-based off-manifold proxy in a narrow setting. This supports the claim as a plausible interpretation, but not yet as a well-established mechanism across methods, models, and tasks. In particular, the paper does not show that the same phenomenon consistently explains ICD, CODE, HalTrapper, or other recent decoding methods.

2.  **The evaluation scope is too narrow for the paper's claims.** The empirical study uses only two backbones and two hallucination benchmarks. For a paper making broad claims about trustworthy MLLM decoding, this is a limited test bed. Recent works in this area often report broader evaluation suites, including combinations of POPE, AMBER, MMHal-Bench or related capability-retention benchmarks. The current evidence is therefore not strong enough to establish generality.

3.  **The most important design choices are insufficiently ablated.** The paper varies `alpha` and `lambda`, but the more consequential decisions are the construction of the prior subspace itself: the blind prompt pool, the use of null images, the choice of final-layer hidden states only, the rank `K=5`, and the sample size `N=50`. Those choices are central to the method, yet the paper provides little evidence that the approach is robust to them.

4.  **The related-work coverage is incomplete.** This weakens both the presentation and the originality case. In particular, the paper should discuss additional related inference-time hallucination-mitigation work that is especially close in spirit, such as **Activation Steering Decoding** (ACL 2025), **Dynamic Correction Decoding / DeCo** (ICLR 2025), and **Mixture of Decoding / MoD** (Findings of ACL 2025). These are highly relevant because they also intervene at decoding time through hidden-state manipulation, prior-aware correction, or adaptive strategy selection. Without discussing them, the paper's novelty is not positioned clearly enough.

---

> ### Author Rebuttal · Authors · 2026-03-28
>
> We sincerely thank the reviewer for the constructive questions.
>
> ### Q1. Difference from ASD / DeCo / MoD, and why MGAP is conceptually distinct.
>
> MGAP  performs **a single-branch, representation-level correction before logits are formed**, rather than multi-branch logit-level correction.
>
> **As defined in Eq. (1), decoding follows "hidden state → decoder head → logits → softmax".** ASD / DeCo / MoD rely on contrastive or logit-level manipulation. In contrast, MGAP intervenes earlier in representation space via a bounded subspace correction that removes harmful priors and preserves orthogonal semantics.
>
> Besides, MGAP uses only a single forward pass per step, while contrastive baselines rely on multiple branches or extra output-level comparisons. As already shown in Table 5, this yields nearly a 2× speedup over contrastive decoding baselines .
>
> ### Q2. Stronger evidence that manifold departure is a general mechanism.
>
> We conduct experiments on 3000 POPE samples, and the results are shown in Table x1.
>
> Interestingly, methods with higher off-manifold ratios consistently perform worse in Table 1 on LLaVA-v1.5-7B. This interpretation is also consistent with prior geometric analyses such as [1], which show that off-manifold perturbations tend to be unreliable. Notably, at α=0.25 MGAP can even reduce the off-manifold ratio relative to vanilla decoding.
>
> **Table x1: Number of off-manifold samples (out of 3000) under different extrapolation coefficients α. Off-manifold samples are defined as those with kNN distance $d_k > \tau$ (as in Figure 3).**
> | Method      | α = 0 | α = 0.25 | α = 0.5 | α = 0.75 | α = 1.0 |
> |------------|------|----------|--------|----------|--------|
> | Vanilla     | 150  | —        | —      | —        | —      |
> | VCD         | 150  | 515      | 2589   | 2999     | 3000   |
> | ICD         | 150  | 416      | 679    | 1175     | 1887   |
> | CODE        | 150  | 775      | 1642   | 2403     | 2944   |
> | HalTrapper  | 150  | 2098     | 2672   | 2918     | 2989   |
> | MGAP    | 150  | 76   | 210| 285  | 402|
>
>
>
> ### Q3. Sensitivity to prior-subspace construction.
>
> MGAP is stable with respect to prompt pool, N, and K.
>
> **We use a blind prompt pool to construct the prior subspace.** On POPE, the blind prompts are templated object-probing questions, roughly of the form “Is there a [object] in the image?”.  On CHAIR, the prompt is fixed as “Please help me describe the image in detail.”, so varying the blind prompt pool is effectively inactive and the metrics remain unchanged.
>
> We additionally conducted ablations on N and K, and the results show that MGAP is robust to both choices in Table x2.
>
> We use the final-layer hidden state as the intervention location. This is because  choosing intermediate layers could introduce unfair layer-specific advantages [2].
>
> We use null-image to construct the prior subspace. This provides a clean way to isolate the language-prior component while minimizing visual evidence. We will clarify these design motivations and limitations more explicitly in the revision.
>
> **Table x2.Sensitivity analysis on POPE (LLaVA v1.5 7B). R/P/A = Random / Popular / Adversarial.**
> | Setting | Value | Acc (R/P/A) | Prec (R/P/A) | F1 (R/P/A) |
> |---|---:|---|---|---|
> | K | 1   | 90.60 / 87.87 / 82.70 | 92.89 / 87.82 / 79.64 | 89.34 / **87.87** / 84.35 |
> | K | 3   | 90.53 / **88.80** / 84.10 | **93.87** / **91.62** / **85.49** | 90.16 / 87.73 / 83.69 |
> | K | 5   | **90.63** / 88.10 / **84.59** | 93.69 / 91.50 / 85.06 | **90.29** / 87.59 / **84.46** |
> | K | 10  | **90.63** / 87.77 / 82.97 | 93.38 / 87.99 / 84.24 | 90.13 / 87.73 / 83.70 |
> | Setting | Value | Acc (R/P/A) | Prec (R/P/A) | F1 (R/P/A) |
> | N | 10  | 90.40 / 87.93 / 83.43 | 93.10 / 89.30 / 81.68 | 89.98 / **87.72** / 83.38 |
> | N | 50  | **90.63** / 88.10 / **84.59** | 93.69 / **91.50** / 85.06 | **90.29** / 87.59 / **84.46** |
> | N | 100 | 90.37 / 88.83 / 84.20 | **94.10** / 91.16 / **85.36** | 89.94 / 87.62 / 83.68 |
> | N | 200 | 90.33 / **88.95** / 84.17 | 94.03 / 91.27 / 85.31 | 89.91 / 87.56 / 83.65 |
>
> ### Q4. AMBER benchmark.
> Yes , we added experiments on the broader hallucination benchmark AMBER, where MGAP remains state-of-the-art.
>
> **Table x3.Results on AMBER (LLaVA v1.5 7B).**
> | AMBER | Model            | CHAIR ↓ | Cover ↑ | Hal ↓ | Cog ↓ |
> |------|------------------|--------|--------|------|------|
> |      | + LLaVa v1.5 7B    | 11.2   | 50.2   | 47.9 | 4.6  |
> |      | + VCD              | 8.9    | 51.2   | 38.1 | 4.4  |
> |      | + ICD              | 8.6    | 51.1   | 37.3 | 3.9  |
> |      | + CODE             | 9.0    | 51.1   | 39.5 | 4.3  |
> |      | + Haltrapper       | 8.0    | 51.5   | 36.3 | 3.8  |
> |      | **+ Ours**         | **7.6**| **51.7**| **35.1** | **3.8** |
>
> [1] Stutz, D. Disentangling Adversarial Robustness and Generalization. CVPR, 2019.
>
> [2] Li, Z. The Hidden Life of Tokens: Reducing Hallucination of Large Vision-Language Models via Visual Information Steering. ICML, 2025.

---

> > ### Author Rebuttal · Reviewer_E9hf · 2026-04-03
> >
> > Thank you for the detailed rebuttal. My concerns are partially resolved. I appreciate the added sensitivity analysis on $K$ and $N$, the clarification of the blind prompt pool and null-image construction, and the additional AMBER results, which strengthen the empirical support for the method.
> >
> > However, my main concerns are only partially addressed. The rebuttal makes the manifold-departure explanation more plausible, but it is still not fully established as a general mechanism across models and tasks (as the new analysis is limited to LLaVA-1.5 on POPE). In addition, while the authors clarify the conceptual differences between MGAP and closely related decoding-time methods (such as ASD, DeCo, and MoD), the empirical comparison and overall positioning relative to these methods remain somewhat limited.
> >
> > My follow-up questions are:
> >
> > 1. Could the authors clarify whether the off-manifold mechanism and performance trends can be shown consistently on the other backbone evaluated in the paper, and ideally on a benchmark beyond the current POPE setting?
> >
> > 2. While I understand MGAP is conceptually distinct from ASD, DeCo, and MoD, could the authors provide an empirical comparison against at least one or two of these closely related inference-time methods to demonstrate that MGAP remains competitive in practice?

---

> > > ### Author Response · Authors · 2026-04-05
> > >
> > > Thank you for these important suggestions. We hope the newly added experiments and clarifications now fully address your main concerns and better demonstrate both the empirical competitiveness and the scope of MGAP.
> > > ### A1:
> > > **We have now extended the same off-manifold analysis to the other backbone in the paper, Qwen3-VL-8B,** and observe the same overall trend: stronger uniform suppression leads to much larger off-manifold growth, while MGAP remains substantially more stable. This strengthens our mechanism claim across both evaluated backbones.
> > >
> > > We agree that analysis beyond POPE would be valuable. We did not include it in the rebuttal because **the off-manifold proxy is much cleaner on POPE, where each example is a yes/no question and the statistic can be computed from the hidden state of a single answer token.** In contrast, descriptive multi-token benchmarks introduce a nontrivial additional design choice—which token(s) to measure and how to aggregate them fairly across variable-length outputs. We therefore prioritized the cross-backbone extension first, and will clarify this limitation in the revision.
> > >
> > > **Table R1. Number of off-manifold samples (out of 3000) under different extrapolation coefficients α on Qwen3-VL-8B and LLaVa v1.5 7B. Off-manifold samples are defined as those with kNN distance $d_k > \tau$
> > >  (as in Figure 3 in our paper).**
> > > | Backbone | Method | α = 0 | α = 0.25 | α = 0.5 | α = 0.75 | α = 1.0 |
> > > |---|---|---:|---:|---:|---:|---:|
> > > | LLaVA v1.5 7B | Vanilla | 150 | — | — | — | — |
> > > |  | VCD | 150 | 515 | 2589 | 2999 | 3000 |
> > > |  | ICD | 150 | 416 | 679 | 1175 | 1887 |
> > > |  | CODE | 150 | 775 | 1642 | 2403 | 2944 |
> > > |  | HalTrapper | 150 | 2098 | 2672 | 2918 | 2989 |
> > > |  | MGAP | 150 | 76 | 210 | 285 | 402 |
> > > | Qwen3-VL 8B | Vanilla | 121 | — | — | — | — |
> > > |  | VCD | 121 | 448 | 2386 | 2979 | 3000 |
> > > |  | ICD | 121 | 362 | 694 | 1187 | 1804 |
> > > |  | CODE | 121 | 724 | 1548 | 2297 | 2886 |
> > > |  | HalTrapper | 121 | 1961 | 2604 | 2881 | 2970 |
> > > |  | MGAP | 121 | 143 | 236 | 321 | 438 |
> > >
> > > ### A2:
> > > **We have additionally included DeCo and MoD results on POPE for both LLaVA v1.5-7B and Qwen3-VL-8B.** These two new baselines already provide a meaningful empirical comparison against recent closely related inference-time methods, and they strengthen the practical positioning of MGAP beyond the original rebuttal.
> > >
> > > We did not add ASD mainly for an objective reproducibility reason: during the rebuttal period, **we were unable to identify a stable official public implementation of ASD.** In contrast, DeCo and MoD both provide public codebases. Moreover, adapting decoding-time methods to our backbones—especially Qwen3-VL-8B—requires nontrivial model-specific engineering. We therefore prioritized the two methods that could be reliably reproduced within the rebuttal window, and will further expand the discussion of ASD in the revision.
> > >
> > >
> > > **Table R2. Empirical comparison with newly added recent baselines (DeCo and MoD) on POPE**.
> > > | Setting | Value | Acc (R/P/A) | Prec (R/P/A) | F1 (R/P/A) |
> > > |---|---|---|---|---|
> > > | LLaVA v1.5 7B | Vanilla | 88.88 / 86.23 / 80.16 | 89.70 / 83.28 / 74.93 | *89.76* / 86.80 / 82.05 |
> > > |  | VCD | 87.57 / 84.23 / 78.56 | 86.23 / 80.76 / 73.47 | 87.79 / 85.08 / 80.66 |
> > > |  | ICD | 89.47 / 87.50 / 82.70 | *92.84* / 89.04 / 81.10 | 89.04 / 87.24 / 83.13 |
> > > |  | HalTrapper | 88.67 / 85.40 / 80.30 | 88.77 / 83.31 / 76.02 | 88.65 / 85.84 / 81.80 |
> > > |  | DeCo (new) | *89.86* / *87.72* / *83.18* | 92.41 / *89.36* / *82.47* | 89.58 / *87.31* / *83.71* |
> > > |  | MoD (new)| 89.24 / 87.03 / 82.51 | 91.37 / 87.96 / 80.74 | 89.17 / 86.91 / 83.02 |
> > > |  | Ours | **90.63** / **88.10** / **84.59** | **93.69** / **91.50** / **85.06** | **90.29** / **87.59** / **84.46** |
> > > | Qwen3-VL 8B | Vanilla | 91.53 / 84.20 / 80.00 | 89.08 / 78.28 / 73.20 | 91.79 / 85.70 / 82.56 |
> > > |  | VCD | 90.67 / 83.27 / 81.36 | 88.80 / 79.41 / 74.92 | 90.83 / 84.53 / 83.60 |
> > > |  | ICD | 91.74 / 84.80 / 82.52 | 91.21 / 80.63 / 76.53 | 91.75 / 85.90 / *83.89* |
> > > |  | HalTrapper | 91.60 / 85.94 / 82.14 | *92.51* / 82.20 / 78.45 | 91.93 / 86.06 / 82.57 |
> > > |  | DeCo (new)| **91.96** / *86.01* / *82.74* | 92.36 / *83.02* / *79.18* | **91.98** / *86.29* / 83.81 |
> > > |  | MoD (new)| 91.87 / 85.76 / 82.61 | 91.94 / 82.71 / 78.92 | 91.86 / 86.12 / 83.74 |
> > > |  | Ours | *91.83* / **86.40** / **83.13** | **93.67** / **84.13** / **79.30** | *91.94* / **86.84** / **84.17** |

---

### Official Review · Reviewer_uD7f · 2026-03-13

**Soundness:** 3
**Presentation:** 3
**Significance:** 2
**Originality:** 3
**Overall Recommendation:** 5
**Confidence:** 3

**Summary:**

This paper investigates the issue of object hallucination during the decoding process of multimodal large language models (MLLMs). It points out that traditional global contrastive decoding methods (e.g., VCD) disrupt the model's internal representations, leading to "Manifold Departure." To address this, the authors propose a training-free intervention method called MGAP. This method offline extracts a "linguistic prior subspace" and, during online decoding, adaptively projects the hidden states by combining the model's uncertainty and image-text consistency. Experiments show that this method outperforms existing contrastive decoding baselines on the POPE and CHAIR datasets, with significantly reduced inference latency.

**Compliance With Llm Reviewing Policy:**

Affirmed.

**Final Justification:**

My concerns are fully solved. I would like to keep the original rating.

**Key Questions For Authors:**

1.  Could ablation experiments on the subspace dimension $K$ be provided? Specifically, how do different values of $K$ (e.g., 1 or 10) affect the final F1 score and CHAIR metrics?
2.  What specific content is included in the 50 prompts used to construct the subspace in Section 4.1? If another set of queries with completely different distributions (e.g., all medical descriptions or all code instructions) were used, would the learned prior subspace $V_{prior}$ change significantly? Would this affect the generalizability of the method?

**Limitations:**

No.
The authors only briefly mention concerns about robustness in the "Impact Statement" at the end of the paper, but no dedicated discussion of limitations is included in the main text or appendix. It is recommended that the authors explicitly discuss under what extreme scenarios the method might fail (e.g., when the visual features of an image are deeply entangled with extremely strong linguistic priors, whether simple linear subspace projection can still effectively strip away erroneous features).

**Strengths And Weaknesses:**

**Strengths:**
*   From the perspective of geometric manifolds, the explanation of why traditional contrastive interventions fail is persuasive. The theoretical proof (Appendix A) and PCA visualization (Figure 3) provide solid support for the method.
*   The article has a compact structure, with a logical and smooth progression from "phenomenon discovery (Figure 2/Figure 3)" to "method proposal." The mathematical expressions are clear.
*   It has high practical value. Compared to methods like VCD and ICD that require multiple forward passes, MGAP completes the intervention in a single forward pass, reducing inference time by nearly 50% (Table 5), making it highly suitable for practical deployment.
*   Abandoning global penalty at the logits level and instead performing orthogonal preservation and intervention in the hidden state subspace is a novel and highly inspiring perspective.

**Weaknesses:**
*   The experimental section lacks necessary ablation studies on key hyperparameters. For instance, how was the principal component dimension $K=5$ determined when constructing the subspace offline? No sensitivity analysis for this parameter is provided.
*   Some implementation details are missing. For example, Section 4.1 mentions using 50 text queries ($N=50$) to extract blind hidden states but does not specify the specific content, length distribution, or domain source of these prompts in the main text or appendix.
*   The authors only briefly mention concerns about robustness in the "Impact Statement" at the end of the paper, but no dedicated discussion of limitations is included in the main text or appendix. It is recommended that the authors explicitly discuss under what extreme scenarios the method might fail (e.g., when the visual features of an image are deeply entangled with extremely strong linguistic priors, whether simple linear subspace projection can still effectively strip away erroneous features).

---

> ### Author Rebuttal · Authors · 2026-03-28
>
> We sincerely thank the reviewer for the positive assessment and the constructive questions on prior-subspace construction.
>
> ### Q1. Sensitivity to the subspace dimension K.
>
> **MGAP is robust to moderate choices of K, and our default choice K=5 lies in a stable high-performance region.**
>
> We added sensitivity experiments on K and N on POPE. We find that MGAP is stable for moderate ranks, with the best overall trade-off around K=3∼5. In particular, K=1 underfits the dominant prior directions, while K=10 slightly weakens selectivity and causes mild degradation on some splits. This supports our default choice of K=5 as a robust middle ground rather than a narrowly tuned value.
>
> For CHAIR, the results are identical across different choices of N and K. This is because CHAIR uses a fixed prompt (“Please help me describe the image in detail.”), so varying N or K does not change the construction in practice. We therefore omit these unchanged results here for brevity.
>
> **Table 1.Sensitivity analysis on POPE (LLaVA v1.5 7B). R/P/A = Random / Popular / Adversarial.**
>
> | Setting | Value | Acc (R/P/A) | Prec (R/P/A) | F1 (R/P/A) |
> |---|---:|---|---|---|
> | K | 1   | 90.60 / 87.87 / 82.70 | 92.89 / 87.82 / 79.64 | 89.34 / **87.87** / 84.35 |
> | K | 3   | 90.53 / **88.80** / 84.10 | **93.87** / **91.62** / **85.49** | 90.16 / 87.73 / 83.69 |
> | K | 5   | **90.63** / 88.10 / **84.59** | 93.69 / 91.50 / 85.06 | **90.29** / 87.59 / **84.46** |
> | K | 10  | **90.63** / 87.77 / 82.97 | 93.38 / 87.99 / 84.24 | 90.13 / 87.73 / 83.70 |
> | Setting | Value | Acc (R/P/A) | Prec (R/P/A) | F1 (R/P/A) |
> | N | 10  | 90.40 / 87.93 / 83.43 | 93.10 / 89.30 / 81.68 | 89.98 / **87.72** / 83.38 |
> | N | 50  | **90.63** / 88.10 / **84.59** | 93.69 / **91.50** / 85.06 | **90.29** / 87.59 / **84.46** |
> | N | 100 | 90.37 / 88.83 / 84.20 | **94.10** / 91.16 / **85.36** | 89.94 / 87.62 / 83.68 |
> | N | 200 | 90.33 / **88.95** / 84.17 | 94.03 / 91.27 / 85.31 | 89.91 / 87.56 / 83.65 |
>
> ### Q2.1. What are the 50 prompts used to construct the subspace?
>
>  The 50 prompts are blind text queries sampled from the task prompt family.
>
> On POPE, the blind prompts are templated object-probing questions, **roughly of the form “Is there a [object] in the image?”**.  These prompts are designed to capture the blind linguistic prior of the task while removing image information.
>
> On CHAIR, the setup uses **a fixed captioning prompt: “Please help me describe the image in detail.”** Therefore, unlike POPE, CHAIR does not rely on a diverse blind prompt set for subspace construction.
>
> ### Q2.2. Why does performance already saturate around N=50?
>
> Because most POPE queries share the same template and differ mainly in the queried object token. **As a result, the induced blind prior subspace is relatively low-dimensional, a pool of around N=50 already captures the dominant blind-prior directions well**. This explains why performance quickly saturates instead of continuing to improve with substantially larger N.
>
> ### Q2.3. Would a very different query distribution change the learned subspace? Would this affect generalizability?
>
> Yes, a substantially different query distribution would likely change the learned prior subspace.
> The prior subspace is shaped by the dominant linguistic patterns of the prompt family, so a very different query distribution would naturally induce a different subspace.
>
> **However, MGAP does not learn a benchmark-specific subspace, but rather a domain-level prior subspace that stays largely stable within the same domain.** For example, we use the same learned subspace for both CHAIR and our additional AMBER experiments, where MGAP still remains state-of-the-art. Therefore, the main sensitivity is expected under cross-domain shifts, rather than within-domain variations.
>
> **Table 2.Results on AMBER (LLaVA v1.5 7B).**
> | AMBER | Model            | CHAIR ↓ | Cover ↑ | Hal ↓ | Cog ↓ |
> |------|------------------|--------|--------|------|------|
> |      | LLaVa v1.5 7B    | 11.2   | 50.2   | 47.9 | 4.6  |
> |      | + VCD              | 8.9    | 51.2   | 38.1 | 4.4  |
> |      | + ICD              | 8.6    | 51.1   | 37.3 | 3.9  |
> |      | + CODE             | 9.0    | 51.1   | 39.5 | 4.3  |
> |      | + Haltrapper       | 8.0    | 51.5   | 36.3 | 3.8  |
> |      | **+ Ours**         | **7.6**| **51.7**| **35.1** | **3.8** |

---

> > ### Author Rebuttal · Reviewer_uD7f · 2026-04-02
> >
> > Thanks for the further clarification. I have no further questions.

---

> > > ### Author Response · Authors · 2026-04-02
> > >
> > > Thank you again for your positive feedback and for the time you've dedicated to reviewing our work. We are happy that our responses met your expectations. **Wishing you all the best!**

---

### Decision · Program_Chairs · 2026-04-30

**Decision:**

Accept (regular)

**Comment:**

This paper received mixed but overall positive feedback (A/WA/WA/WR). Reviewers agree that MGAP is attractive due to attributes of training-free, efficient, and simple forward pass. The rebuttal effectively addressed most of the concerns by investigating sensitivity, clarifying blind prompt and extending the evaluation to AMBER benchmark, etc. One remaining concern focuses on the degradation of VCD may depend on an unusually strong APC setting, which seems to weaken the paper's applicable scope. However, it does not compromise the core contribution of MGAP and its effectiveness across multiple baselines. Overall, I would recommend a Weak Accept, while the paper needs to further address this concern in camera ready.